# Carbon tax acceptability with information provision and mixed revenue uses

Sara Maestre-Andrés [1✉], Stefan Drews[1], Ivan Savin[1,2] & Jeroen van den Bergh[1,3,4]

Public acceptability of carbon taxation depends on its revenue use. Which single or mixed revenue use is most appropriate, and which perceptions of policy effectiveness and fairness explain this, remains unclear. It is, moreover, uncertain how people's prior knowledge about carbon taxation affects policy acceptability. Here we conduct a survey experiment to test how distinct revenue uses, prior knowledge, and information provision about the functioning of carbon taxation affect policy perceptions and acceptability. We show that spending revenues on climate projects maximises acceptability as well as perceived fairness and effectiveness. A mix of different revenue uses is also popular, notably compensating low-income households and funding climate projects. In addition, we find that providing information about carbon taxation increases acceptability for unspecified revenue use and for people with more prior tax knowledge. Furthermore, policy acceptability is more strongly related to perceived fairness than to perceived effectiveness.

[1] Institute of Environmental Science and Technology (ICTA), Universitat Autònoma de Barcelona, Barcelona, Spain. [2] Graduate School of Economics and Management, Ural Federal University, Yekaterinburg, Russian Federation. [3] ICREA, Barcelona, Spain. [4] School of Business and Economics & Institute for Environmental Studies, VU University Amsterdam, Amsterdam, The Netherlands. ✉email: sara.maestre@uab.cat

Carbon taxes are widely seen as a key instrument of climate policy. Although many countries and regions have already implemented them in some form, they still face resistance. Witness the French "yellow vests" protests[1], cancellation of an Australian tax in 2014[2], and rejection of a carbon tax in repeated referenda in Washington State[3]. This underpins the need to better understand public attitudes to carbon taxation. In searching for ways to garner more public support, a central question is how to use tax revenues[4]. So far, three uses dominate: support of climate projects, part of general public funds, and transfers to firms or individuals[5].

A carbon tax tends to have a regressive effect in industrialized countries since poor households spend a relatively large share of their income on carbon-intensive subsistence goods[6]. However, a specific case of the third revenue use type, namely compensating poor households, can ameliorate such an effect[7]. Previous studies show that although many people are concerned about the potentially uneven distribution of the policy burden, they do not show a clear preference for using the carbon tax revenues in ways that reduce any inequitable effects. Instead, use of revenues for climate projects, such as supporting renewable energy, frequently emerges as the most preferred option[8–10]. Other studies find that factors like political context, information provision about expected emissions reduction or about distributional effects can shift support of revenue uses away from climate projects to compensating inequitable effects[11–14]. While revenue allocation in reality often covers multiple purposes[5,13], studies of public attitudes have so far focused on single revenue uses[4,15]. This raises the question how people respond to carbon tax proposals under mixed revenue uses. In view of public concerns about regressive effects, testing this should include revenue uses that address equity issues.

The effect of the revenue uses on public acceptability of carbon taxes is mediated by perceptions of the expected outcomes of the policy, such as its fairness and effectiveness in terms of emissions reduction[10]. When analysing perceived fairness of climate change policies, two main dimensions can be distinguished: individual or 'fairness to me' and distributional or 'fairness to others'[16–18]. The latter is captured by the distribution of effects across individuals or groups.

To what extent revenue use shapes acceptability is likely to depend also on people's prior knowledge and information provision about carbon taxation. In particular, the widespread preference for using tax revenue to support climate projects has been suggested to indicate that people do not understand the regulatory effect of a carbon tax, that is, reduction of emissions regardless of the use of its revenues[7,19]. Some real-world examples of (similar) environmental taxes show that people have revised their beliefs about the effectiveness and perceptions of these policies once the policy has been implemented[20–23]. However, before policy implementation and actual experiences, information provision and communication are crucial to improve public understanding and garner sufficient public support for carbon taxes[24]. Research includes providing people with detailed information on carbon tax effects from computable general equilibrium models[11,12], examining the impact of providing information about emission reductions of carbon taxation according to experts[25,26], or information about the effectiveness of other policies[27].

Against this background, we offer a new approach that examines how single and mixed revenue uses in combination with prior knowledge and information provision about the policy affect acceptability, and whether this can be explained by perceived effectiveness and fairness. To this end, we conducted an online survey experiment among the general public of Spain ($N = 2004$) about their perceptions of, and attitudes to, a carbon tax. No studies about carbon tax perceptions have been previously undertaken for Spain, with the exceptions of Heres et al.[28], who conducted laboratory experiments with students at the University of the Basque Country in Bilbao, and Savin et al.[29], who undertook a computational-linguistic analysis of citizens' associations with a carbon tax and its fairness. There is no genuine carbon tax implemented in Spain, nor is there any serious public debate about this instrument of climate policy at the moment (see Supplementary Note 1 for more detail).

While many of the above-mentioned studies have examined attitudes to existing carbon taxes, it is informative for policymakers to learn about public opinion before implementation, also as this allows a wider variety of policy design options. To test the effect of information provision, half of the sample was given a short text explaining how a carbon tax works (see Supplementary Notes 2 and 3 for a detailed description of the survey questions and information provided, in English and Spanish languages). The data was collected by a professional survey company during August 2019, using quotas on age, gender and geographical regions to achieve a nationally representative sample on these socio-demographic dimensions (for more details, see the 'Methods' section and Supplementary Table 1). We use quotas on age, gender and geographical distribution to ensure similarity among treatment and control groups. In addition, we tested that these samples do not differ on other covariates, such as climate concern and education (see Supplementary Table 2). The survey addressed the following research questions: (a) How do knowledge and information about carbon taxation influence acceptability? (b) Which revenue use maximizes acceptability? And (c) how does acceptability relate to perceived effectiveness and fairness? We further examine the influence of perceptions of individual effects and distributional effects on perceived fairness regarding the carbon tax. Here we address individual effects by asking respondents how they think a carbon tax will affect them personally, on a scale from 'much worse off' to 'much better off' (labelled hereafter as 'personal effects'). We address distributional effects by asking respondents how they think a carbon tax will affect low-income households, on a scale from 'much worse off' to 'much better off' (labelled hereafter as 'low-income effects'). Moreover, we consider five types of revenue uses: (i) support the development of climate projects (Climate); (ii) equal transfers to households (AllHH); (iii) transfers only to low-income households (PoorHH); (iv) half of the revenues to support development of climate projects and the rest as equal transfers to households (AllHH&Climate); and (v) half of the revenues to support the development of climate projects and the rest as transfers to low-income households (PoorHH&Climate).

## Results

**Effects of knowledge and information provision about carbon taxes.** First, we examine how assessed and self-perceived knowledge about a carbon tax affects acceptability and perceptions. To determine assessed knowledge, we asked respondents to answer six questions as "true", "false" or "don't know" (Fig. 1a). Their responses were aggregated using a Mokken scale[30]. In particular, we count only correct responses as an indication of knowledge and exclude the fifth item due to its weak scalability (see 'Methods' section). The resulting indicator is shown in Fig. 1b. We measure self-perceived knowledge by asking respondents to assess their knowledge about a carbon tax on a scale between 1 and 5 (Fig. 1c). Most people perceive themselves to have little or no knowledge at all.

The distribution of assessed and self-perceived knowledge among people with different levels of acceptability of a carbon tax, for unspecified revenue use, is demonstrated in Fig. 2. It

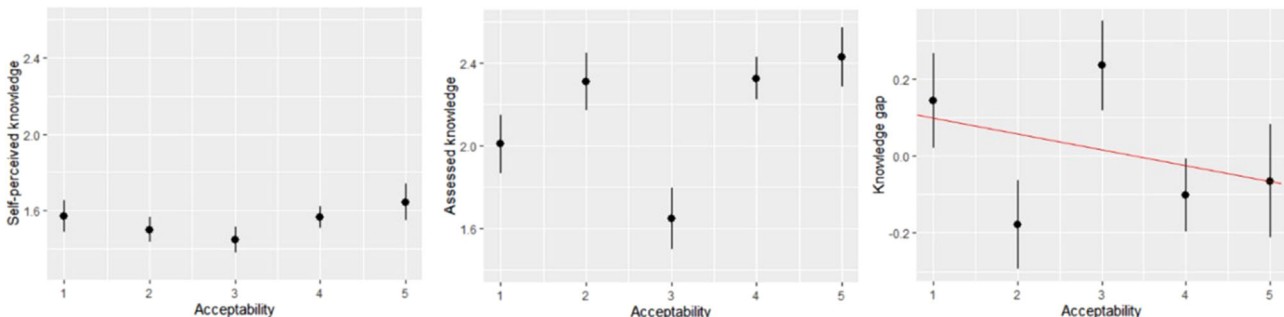

**Fig. 1 Construction of, and relation between, assessed and self-perceived knowledge. a** Items used to assess knowledge about a carbon tax and their response distribution (correct answers in capital letters in brackets); **b** distribution of assessed knowledge; **c** distribution of self-perceived knowledge; **d** rank correlation with a 2D-density plot of self-perceived versus assessed knowledge, illustrating where most observations are concentrated. The rank correlation between assessed and self-perceived knowledge is 0.33 and significant at 1% level. The red line in the right plot indicates a fitted linear model.

**Fig. 2 Assessed and self-perceived knowledge for different levels of carbon-tax acceptability.** Data are presented as mean values with error bars representing +/−2 s.e. Acceptability is measured on a 5-point scale ranging from 1 = completely unacceptable to 5 = completely acceptable. The red line in the right plot indicates a fitted linear model. Subgroups sizes are: $N = 382$ for acceptability level 1, $N = 374$ for acceptability level 2, $N = 387$ for acceptability level 3, $N = 589$ for acceptability level 4, and $N = 272$ for acceptability level 5.

reveals that people's assessed knowledge is higher than self-perceived knowledge, particularly among people with a high acceptability of carbon taxation. This may be because respondents underestimate their knowledge about carbon taxes. This interpretation is consistent with other research on knowledge about climate change[31]. Our results further show that people with a low acceptability perceive themselves as having much knowledge about carbon taxation, i.e. when we compare them with the rest of the sample. Even so, the measure of assessed knowledge does not support this perception. This is confirmed by a formal test assessing the knowledge gap for each respondent, i.e. the difference between self-perceived and assessed knowledge (after z-scoring both measures to make them comparable), as shown in the right plot in Fig. 2, and regressing this on the acceptability of a carbon tax, giving a significant negative relationship ($\beta = -0.0385$, $p$ value $= 0.0479$). This result is in line with the findings of a prior study that for controversial topics such as genetically modified food, and to a lesser extent climate change, extreme opponents have limited understanding of the matter but nevertheless think they know the most[32]. Our results further show that respondents who consider a carbon tax as "neither unacceptable nor acceptable" (acceptability score 3) have on average the lowest assessed knowledge and self-perceived knowledge, though in the latter case the difference with other groups is not very large. It may mean that respondents with the least knowledge about the policy are the least interested in the topic and therefore do not have an opinion about it.

Next, we compare assessed knowledge and tax acceptability between two subsamples of respondents: those who did and did not receive information about how the carbon tax works. The results are shown in Fig. 3. The information explained that the carbon tax is a charge on fossil fuels in proportion to the amount of carbon they contain as this determines how many $CO_2$ emissions result from their combustion, as well as consequences in terms of relative fuel and goods/service prices, and subsequent behavioural responses by producers and consumers. Our results show that people receiving information about the functioning of a carbon tax tend to have a higher probability to accept it, particularly those with already relatively high assessed knowledge, i.e. with levels 3 and 4. The difference between the subsamples is significant at 5 and 10% levels, respectively, according to a Kruskal–Wallis rank-sum test ($p$ values at the bottom of Fig. 3). Providing information did not matter for respondents with absolutely no knowledge about carbon taxes (level 1), which might be due to an incapacity or unwillingness to process external information about the carbon tax. For very high assessed knowledge (level 5) the number of observations is too low to assess any significant difference. Supplementary Table 3 further supports these insights by offering results of an ordered logit regression conducted on the whole sample to examine the interaction between assessed knowledge and information provision for unspecified revenue use.

**Impact of revenue use on perceptions and acceptability.** Next, we test which revenue use(s) (including unspecified use) of a carbon tax maximize its acceptability, how they affect perceptions, and how these relationships depend on whether respondents received information or not. Fig. 4 shows the different outcomes for groups with and without the information-provision treatment. To test statistical differences, we undertook a pairwise Mann–Whitney test for different revenue uses and a Kruskal–Wallis rank-sum test for experiment vs control groups (Supplementary Fig. 1). Three main findings are as follows. First, a carbon tax with all revenues spent on support of climate projects is the most accepted option (average acceptance rate 3.88, while other revenue uses stay below 3.5). The two revenue uses

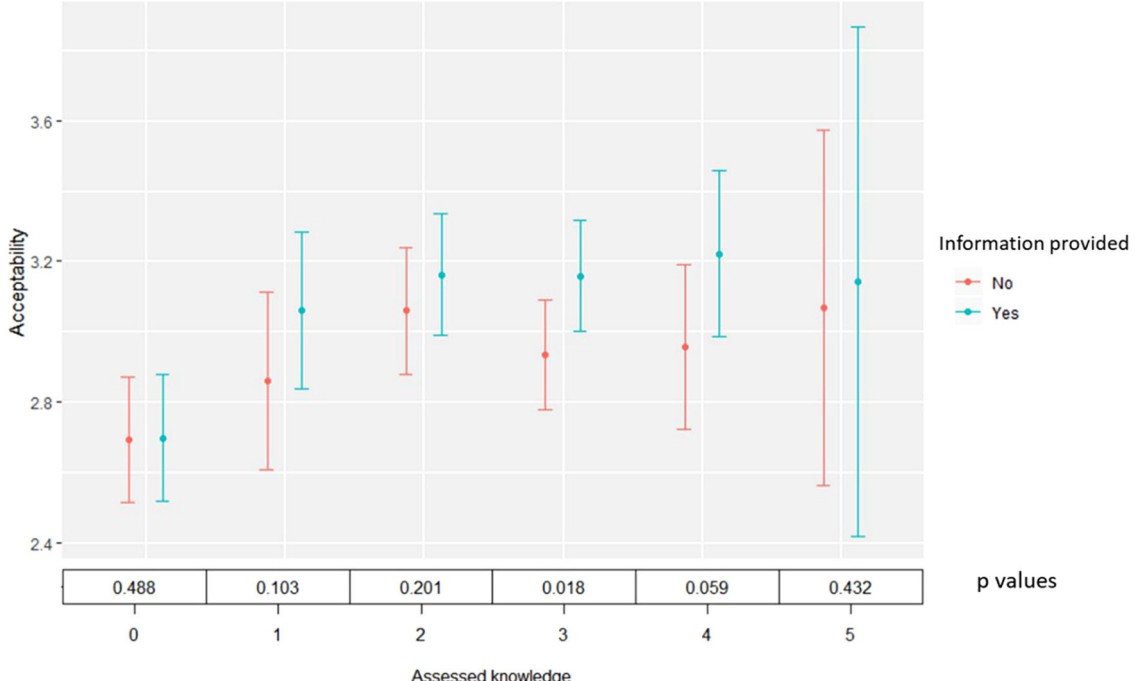

**Fig. 3 Relationship between assessed knowledge and acceptability of a carbon tax with and without information provision, for unspecified revenue use.** Results include averages and error bars depicting $+/-2$ s.e. Acceptability is measured on a 5-point scale: 1 = completely unacceptable to 5 = completely acceptable. At the bottom are $p$-values of a Kruskal–Wallis rank-sum test with Bonferroni correction for statistical differences between the two subsamples. Subgroup sizes are: $N = 353$ for assessed knowledge equalling 0, $N = 256$ for assessed knowledge equalling 1, $N = 488$ for assessed knowledge equalling 2, $N = 600$ for assessed knowledge equalling 3, $N = 264$ for assessed knowledge equalling 4, and $N = 43$ for assessed knowledge equalling 5.

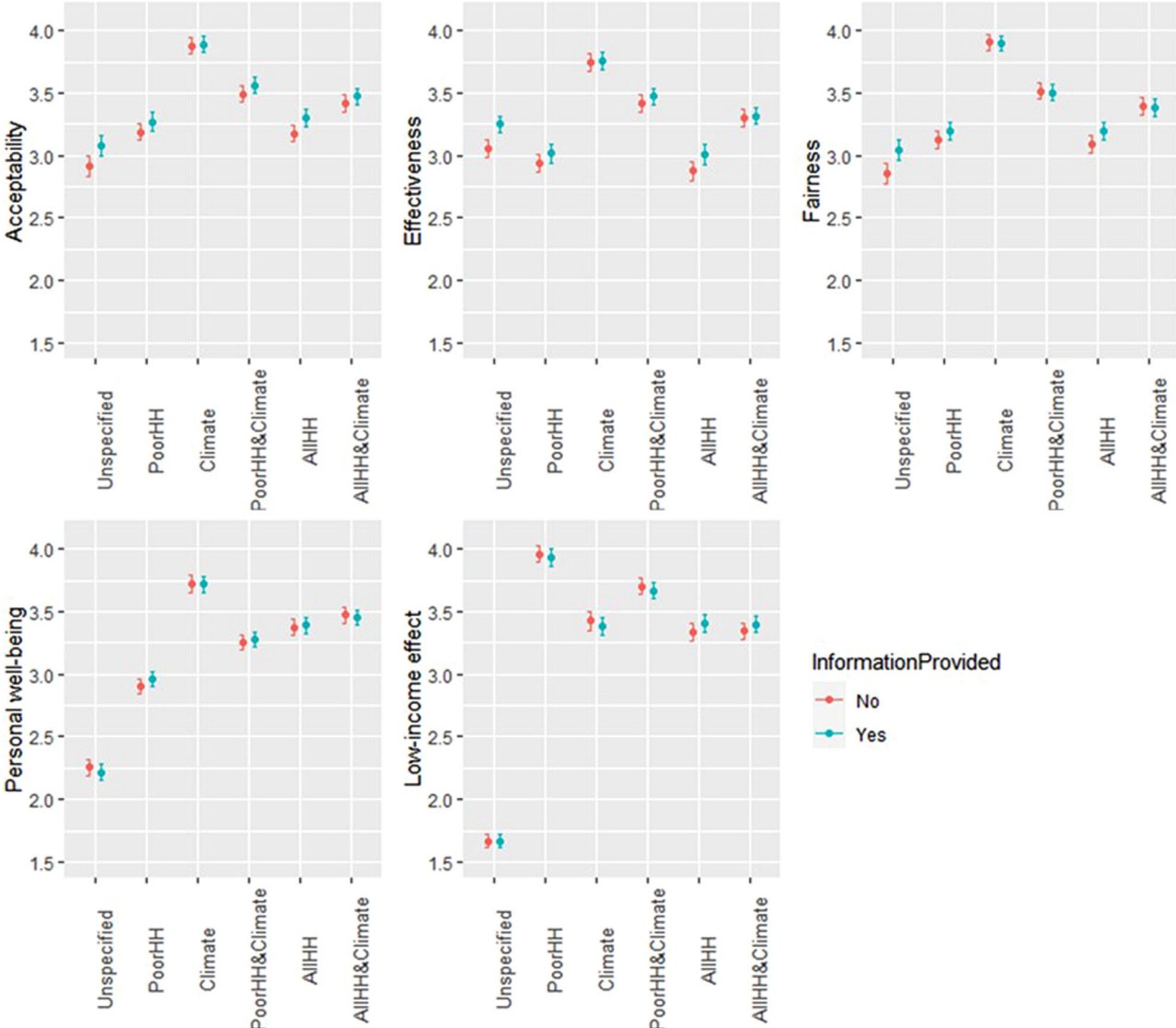

**Fig. 4 Effect of providing information on acceptability and perceptions under distinct revenue uses.** Dots indicate the average results with error bars indicating $+/-2$ s.e. All survey questions were using a 5-point scale (e.g. acceptability: 1 = completely unacceptable to 5 = completely acceptable). $N = 2004$ (1000 with no information provided and 1004 with information provided) for each revenue use option and for acceptability and each of the four perceptions.

that mix climate projects with transfers to either low-income or all households show a higher acceptability than unspecified revenue use and the other two single revenue uses.

Second, using the revenues for supporting climate projects increases perceived effectiveness and fairness compared to both unspecified and other specified uses. The latter result is somewhat surprising given that revenue use for supporting climate projects was perceived by the respondents as relatively unfavourable for poor households. Climate projects were nevertheless considered by respondents to make them better off personally ('personal effects') compared to the other revenue options. The second option perceived as both more effective and fair is distributing the revenues to low-income households and climate projects. Using the revenues for low-income transfers rather than for supporting climate projects was perceived to make poor people better off. Respondents considered returning revenues to poor or all households as the least effective in terms of emissions reduction compared to the other revenue uses.

Third, we find that information provision tends to result in more favourable acceptability, perceived fairness and effectiveness

of a carbon tax for unspecified revenue use and to a lesser extent for revenues going in equal amount to all households, while it shows no significant effect for the other revenue schemes.

We also asked respondents how they would allocate tax revenues to different uses if they had complete freedom to decide about this. The results are summarized in Fig. 5. The left chart shows results for all respondents ($N = 2004$), while the right and central chart present results for respondents who initially accepted a carbon tax ($n = 861$) and those who did not before any revenue use was specified ($n = 756$). We find that more than half of the sample prefers a combination of the three revenue uses: support climate projects, transfers to low-income households and transfers to all households. These respondents generally prefer to allocate a greater share of revenues to support climate projects than to the other two uses, particularly among those respondents accepting a carbon tax (i.e. 45% on average, 41% for those not accepting a carbon tax, and 50% for those accepting). Similarly, a higher preference for spending money on climate projects is found among people accepting the policy but preferring single or mixes of two revenue uses. This generalises

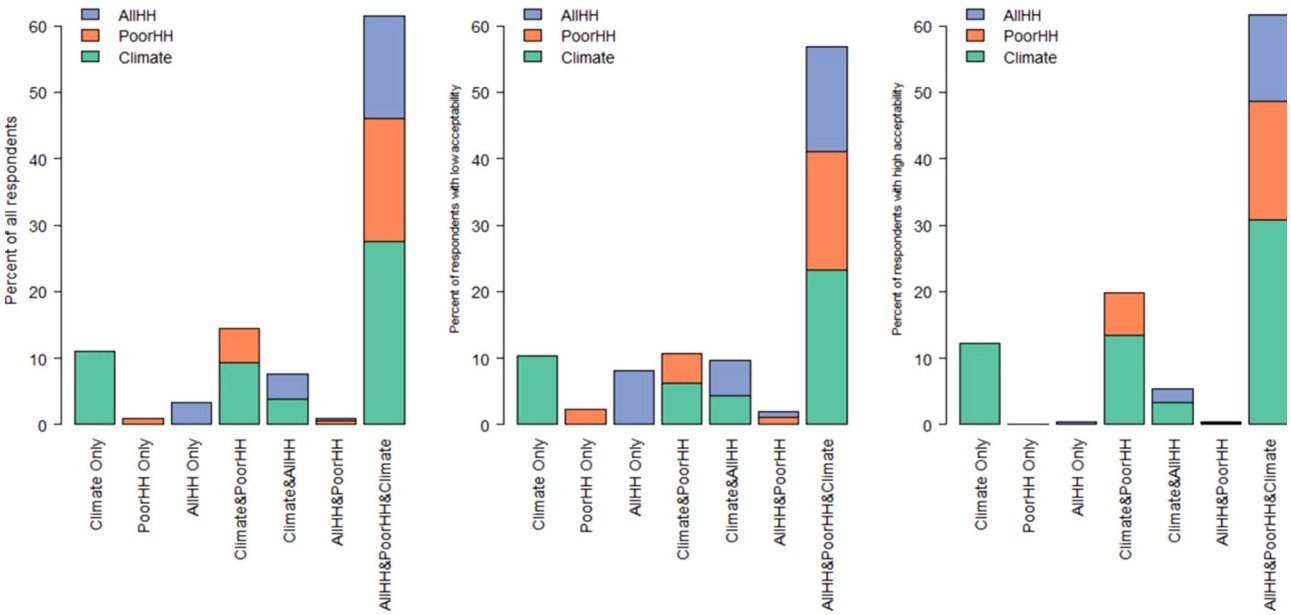

**Fig. 5 Preferred allocation of revenues from a carbon tax.** Respondents were asked to express the share (in %) of total revenues to be allocated to each revenue use. The left chart is for all respondents, while the right and central charts are for respondents who initially accepted or did not accept a carbon tax before any revenue use was specified. The coloured parts of each bar indicate average shares. Respondents with high (low) acceptability are defined as having a response of higher (lower) than 3 (on a scale from 1 to 5). Respondents who chose the median (3) option were excluded from the centre and right plots of the figure.

previous findings in the literature for single revenue uses to mixed revenue uses[11]. On the other hand, opponents of a carbon tax tend to favour transfers to all or low-income households.

We repeated the analysis for indifferent respondents only (i.e. 387 persons who chose the median (3) option when asked about their acceptability of a carbon tax) and find that the distribution of revenues for this group is very similar to that of the total population (Supplementary Fig. 2). We further analysed whether the allocation of revenues for climate projects varied with the acceptability of the policy among respondents who endorsed all three revenue options. Our results show that those who rejected a carbon tax allocated on average 40.74% to climate projects, those indifferent 41.42% and those who supported the tax 49.79%. Applying a Kruskal–Wallis rank-sum test to compare the allocation of revenues to climate projects between these three groups, we find a chi-squared = 58.08 with df = 2 and $p$-value = 0. This means that the allocation of revenues significantly differs between the three groups. More specifically, while choices by those who reject and by those who are indifferent about a carbon tax are not significantly different ($p$-value = 0.82), choices by carbon-tax supporters are significantly different from both those who reject ($p$-value<0.0001) and those who are indifferent ($p$-value<0.0001). Our results thus provide evidence that, despite respondents' fairly low knowledge about carbon taxes (Fig. 1), they do not endorse all possible options equally, as may be expected by the 'ignorance prior' effect (Supplementary Figs. 3 and 4). This effect suggests that respondents prefer all options almost equally due to being ignorant about the topic they are assessing[33].

Next, to examine how carbon tax acceptability and perceptions relate to revenue uses, knowledge and information provision, we ran several ordered logit regression models. These treat perceptions of survey respondents under each of the six revenue uses as separate observations (including the unspecified option). Results are shown in Table 1. Apart from the five specified revenue uses (the unspecified use serves as a benchmark), the models include self-perceived knowledge, interaction terms between assessed

knowledge and the revenue uses (to see if people with higher prior knowledge about the policy tend to prefer any of the revenue uses), and between information and revenue uses (to test if the information provision is improving policy perception under distinct revenue uses). In addition, we account for socio-demographic and other control variables collected in our survey, such as age, gender, education, trust in politicians, climate change concern and political orientation.

Table 1 shows that, in line with Fig. 4, all specified revenue uses significantly increase acceptability compared to unspecified revenue use, with climate projects being the revenue use that increases acceptability and perceived effectiveness and fairness the most.

Information provision is found to affect policy acceptability and perceptions. The results in Table 1 indicate that people receiving information are more likely to accept a carbon tax with unspecified revenue use. In this case, information provision has a positive effect on perceived fairness and effectiveness. This effect is only weakly significant for revenues distributed to all households. For other revenue uses we also observe positive effects, but they are not statistically significant.

One interpretation of the foregoing results is that the information provided makes people aware that a carbon tax, regardless of its revenue use, is effective and fair, raising policy acceptability. This is because people better understand that the regulatory or incentive effect of carbon taxation implies shifts in choices by consumers and firms from high- to low-carbon options. Our results show, therefore, that providing information about carbon tax functioning and specifying its revenue use can achieve a similar effect on acceptability, but when combined lead to a smaller overall effect.

Assessed knowledge about carbon taxation, in contrast, tends to interact most positively with support of climate projects and to a lesser extent with transfers going partly to low-income households and support of climate projects. In other words, the more people know about carbon taxes, the higher is their acceptability for these two uses of revenues. The interaction effect

**Table 1 Determinants of acceptability and perceptions.**

| Explanatory variables | Acceptability | Perceptions | | | |
| --- | --- | --- | --- | --- | --- |
| | | Effectiveness | Fairness | Low-income effects | Personal effects |
| Revenue uses | | | | | |
| PoorHH | 2.86*** (1.93-4.21) | 1.21 (0.83-1.77) | 1.97*** (1.34-2.90) | 32.24*** (21.32-48.85) | 4.54*** (3.08-6.68) |
| Climate | 3.57*** (2.41-5.29) | 2.22*** (1.52-3.26) | 3.13*** (2.12-4.63) | 15.47*** (10.35-23.15) | 9.76*** (6.58-14.50) |
| AllHH | 2.58*** (1.76-3.80) | 1.53** (1.05-2.23) | 2.14*** (1.46-3.15) | 23.52*** (15.76-35.20) | 5.39*** (3.66-7.94) |
| PoorHH&Climate | 3.36*** (2.28-4.96) | 1.53*** (1.05-2.22) | 2.33*** (1.58-3.44) | 19.64*** (13.16-29.39) | 8.09*** (5.48-11.97) |
| AllHH&Climate | 3.07*** (2.08-4.52) | 1.38* (0.95-2.00) | 2.4*** (1.63-3.54) | 16.08*** (10.82-23.94) | 8.28*** (5.61-12.24) |
| Interactions between Assessed knowledge and Revenue uses | | | | | |
| Assessed knowledge*Unspecified | 1.19*** (1.10-1.28) | 1.03 (0.96-1.11) | 1.03 (0.96-1.11) | 1.02 (0.95-1.10) | 0.97 (0.90-1.04) |
| Assessed knowledge*PoorHH | 0.93** (0.87-1.00) | 0.88*** (0.82-0.95) | 0.91** (0.85-0.98) | 1.23*** (1.14-1.33) | 0.89*** (0.83-0.96) |
| Assessed knowledge*Climate | 1.31*** (1.22-1.42) | 1.20*** (1.11-1.29) | 1.25*** (1.17-1.35) | 1.09*** (1.01-1.17) | 1.20*** (1.11-1.29) |
| Assessed knowledge*PoorHH&Climate | 1.13*** (1.05-1.21) | 1.10*** (1.02-1.18) | 1.10** (1.03-1.18) | 1.12*** (1.04-1.21) | 1.06 (0.99-1.14) |
| Assessed knowledge*AllHH | 0.88*** (0.82-0.94) | 0.79*** (0.73-0.85) | 0.85*** (0.79-0.91) | 0.98 (0.91-1.05) | 0.99 (0.92-1.06) |
| Assessed knowledge*AllHH&Climate | 1.03 (0.96-1.11) | 1.03 (0.96-1.10) | 0.98 (0.91-1.05) | 1.03 (0.96-1.11) | 1.06 (0.98-1.14) |
| Interactions between Information provision and Revenue uses | | | | | |
| Information*Unspecified | 1.40*** (1.15-1.70) | 1.30*** (1.08-1.56) | 1.36*** (1.12-1.66) | 0.87 (0.72-1.06) | 0.88 (0.73-1.07) |
| Information*PoorHH | 1.10 (0.92-1.33) | 1.11 (0.92-1.34) | 1.09 (0.91-1.31) | 0.89 (0.73-1.09) | 1.02 (0.84-1.23) |
| Information*Climate | 1.03 (0.85-1.25) | 1.00 (0.83-1.22) | 0.97 (0.80-1.17) | 0.94 (0.78-1.14) | 1.01 (0.83-1.23) |
| Information*PoorHH&Climate | 1.12 (0.93-1.34) | 1.01 (0.83-1.22) | 0.95 (0.79-1.14) | 0.89 (0.74-1.08) | 1.01 (0.84-1.22) |
| Information*AllHH | 1.18* (0.98-1.42) | 1.23** (1.01-1.49) | 1.14** (0.94-1.37) | 1.03 (0.86-1.25) | 1.01 (0.84-1.22) |
| Information*AllHH&Climate | 1.04 (0.87-1.26) | 1.10 (0.91-1.33) | 1.04 (0.87-1.26) | 1.05 (0.87-1.27) | 1.01 (0.84-1.22) |
| Control variables | | | | | |
| Self-perceived knowledge | 0.86*** (0.81-0.91) | 0.91*** (0.86-0.97) | 0.86*** (0.82-0.92) | 0.90*** (0.85-0.96) | 0.89*** (0.84-0.94) |
| Age | 1.00* (0.99-1.00) | 0.99*** (0.99-1.00) | 0.99*** (0.99-1.00) | 1.00** (1.00-1.01) | 1.00 (0.99-1.00) |
| Gender | 1.13*** (1.04-1.22) | 1.17*** (1.07-1.27) | 1.14*** (1.05-1.24) | 1.17*** (1.08-1.28) | 1.12*** (1.03-1.22) |
| Education | 0.97* (0.93-1.01) | 0.94*** (0.90-0.97) | 0.94*** (0.91-0.98) | 0.97 (0.93-1.01) | 0.97*** (0.93-1.00) |
| Climate concern | 1.30*** (1.26-1.35) | 1.16*** (1.12-1.20) | 1.24*** (1.20-1.29) | 1.30*** (1.26-1.35) | 1.29*** (1.25-1.34) |
| Political orientation | 0.91*** (0.89-0.93) | 0.94*** (0.92-0.96) | 0.93*** (0.92-0.95) | 0.97*** (0.96-0.99) | 0.93*** (0.91-0.95) |
| Monthly income | 0.97 (0.94-1.01) | 0.99 (0.95-1.03) | 0.96 (0.91-0.99) | 0.99 (0.95-1.03) | 0.97 (0.93-1.01) |
| Car use | 0.93*** (0.91-0.96) | 0.96*** (0.94-0.99) | 0.96*** (0.94-0.98) | 0.94*** (0.92-0.96) | 0.92*** (0.89-0.94) |
| Trust in politicians | 1.29*** (1.23-1.36) | 1.28*** (1.22-1.35) | 1.22*** (1.16-1.28) | 1.23*** 1.17-1.29 | 1.29*** (1.23-1.35) |
| Household size | 1.02 (0.99-1.06) | 1.02 (0.99-1.06) | 1.01* (0.97-1.04) | 1.01 (0.97-1.04) | 0.99 (0.97-1.03) |
| Nagelkerke pseudo $R^2$ | 0.76 | 0.76 | 0.76 | 0.83 | 0.79 |

Note: Coefficients indicate odds ratios with 2.5-97.5% confidence intervals expressed within brackets. Asterisks ***, ** and * indicate 1%, 5% and 10% significance, respectively.

is negative, however, with transfers to all households and low-income households. A possible explanation is that those with higher assessed knowledge tend to have higher climate concern and prefer to spend money on climate projects.

The results regarding perceptions of how the carbon tax will affect low-income households (penultimate column) reveal that, not surprisingly, transferring the revenues to low-income households is the revenue use that would make low-income households best off, followed by revenues going partly to low-income households and support of climate projects. Providing information about the functioning of the policy does not change the perception of regressive effects. The results for personal effects (final column) show that the use of the revenues to support climate projects is regarded as making people better off personally. Supplementary Note 4 provides some discussion about the influence of socio-demographic variables on policy acceptability and perceptions.

**Relations between perceptions and carbon tax acceptability**. We now assess the connection between the acceptability of a carbon tax and perceptions about its effectiveness, fairness, personal effects and impact on low-income households. According to the results in the second column of Table 2, perceived fairness has a strong positive relation with perceived personal effects and a weaker positive relation with perceived low-income effects. Acceptability is roughly equally positively related to effectiveness and personal effects and less so to low-income effects, as the third column shows. Hence, low-income effects have the weakest relation with both fairness and acceptability. Finally, if we regress acceptability on effectiveness and fairness (as the latter incorporates the role of both, personal effects and low-income effects), the fourth column shows that perceived fairness of a carbon tax is a more important predictor than perceived effectiveness of acceptability. This result is supported by the correlation matrix for the five perception variables (see Supplementary

**Table 2 Determinants of fairness and acceptability.**

| Explanatory variables | Perceived fairness | Acceptability | |
|---|---|---|---|
| **Perceptions** | | | |
| Effectiveness | – | 2.73*** (2.61–2.86) | 1.71*** (1.63–1.80) |
| Fairness | – | – | 4.88*** (4.61–5.18) |
| Personal effects | 2.86*** (2.72–3.00) | 2.26*** (2.15–2.39) | – |
| Low-income effects | 1.19*** (1.14–1.24) | 1.31*** (1.26–1.36) | – |
| **Control variables** | | | |
| Self-perceived knowledge | 0.90*** (0.85–0.95) | 0.91** (0.86–0.97) | 0.92** (0.87–0.98) |
| Assessed knowledge | 1.00 (0.97–1.03) | 1.08*** (1.04–1.11) | 1.09*** (1.06–1.13) |
| Information provided | 1.09*** (1.01–1.18) | 1.12*** (1.04–1.22) | 1.09*** (1.00–1.18) |
| Age | 0.99*** (0.99–1.00) | 1.00 (0.99–1.00) | 0.99 (0.99–1.00) |
| Gender | 1.10* (1.01–1.2) | 0.99 (0.91–1.08) | 0.97 (0.89–1.07) |
| Education | 0.95*** (0.91–0.98) | 1.02 (0.98–1.06) | 1.03 (0.99–1.07) |
| Climate concern | 1.07*** (1.04–1.11) | 1.14*** (1.10–1.19) | 1.18*** (1.14–1.23) |
| Political orientation | 0.96*** (0.94–0.97) | 0.94*** (0.93–0.96) | 0.94*** (0.92–0.96) |
| Monthly income | 0.97* (0.93–1.01) | 0.99 (0.95–1.03) | 1.02 (0.98–1.06) |
| Car use | 1.00 (0.98–1.03) | 0.97 (0.95–0.99) | 0.95*** (0.92–0.97) |
| Trust in politicians | 1.06** (1.01–1.11) | 1.07** (1.01–1.12) | 1.14*** (1.08–1.20) |
| Household size | 1.01 (0.98–1.05) | 1.03 (0.99–1.06) | 1.02 (0.99–1.06) |
| Nagelkerke pseudo $R^2$ | 0.82 | 0.87 | 0.89 |

Note: Coefficients indicate odds ratios with 2.5–97.5% confidence intervals expressed within brackets. Asterisks ***, ** and * denote 1%, 5% and 10% significance, respectively.

Fig. 5) and the rank of the predictors of acceptability by a gradient boosting machines algorithm[34] (see 'Data analysis' section and Supplementary Fig. 6). Incidentally, this is not completely in line with respondents' perceptions as derived from responses to a question directly asking which factor—effectiveness or fairness—played a stronger role in their decision on how to allocate the revenue generated by the carbon tax. On average, people gave slightly more importance to effectiveness (mean value 2.89 on a scale from 1 to 5, with 1 "only effectiveness" and 5 "only fairness"). This may reflect that people sometimes lack conscious awareness of the origin of their attitudes, as confirmed by psychological research on the distinction between explicit and implicit attitudes[35].

This study revealed that a carbon tax with revenues completely devoted to supporting climate projects is the most accepted option and strengthens the perceived effectiveness and fairness of the overall policy. We found that the mixed revenue allocated to low-income households and climate projects has the second-highest acceptability and perceived effectiveness as well as fairness. Nevertheless, when respondents could freely express their preferred revenue allocation, more than half of the sample opted for some combination of the three revenue uses, namely supporting climate projects and transfers to both low-income and all households. On average, though, respondents prefer to allocate a greater share to supporting climate projects than to alternative uses.

Another important finding of our study is that people with more prior knowledge about the policy tend to show a higher acceptability. Furthermore, people who reject the carbon tax think they understand it relatively well, but according to our knowledge assessment they actually know little about it. At the same time, those who accept carbon taxation underestimate their level of knowledge. Overall, we find a negative correlation between people's self-perceived knowledge about carbon taxation and their acceptability of the policy. People with higher assessed knowledge about carbon taxation tend to show a lower acceptability, perceived effectiveness and perceived fairness if revenues are spent on compensating either all or only low-income households. However, their acceptability tends to be higher if revenues are spent on climate projects either in full or partly together with transfers to low-income households.

Providing respondents with information about the functioning of the tax can raise acceptability, effectiveness and fairness when

revenues are unspecified. Hence, information provision about carbon tax functioning and revenue use can both contribute to improving acceptability. However, since their combination has a smaller impact, future research is needed to test whether synergies are possible. Finally, we find a stronger relationship between carbon tax acceptability and perceived fairness than with perceived effectiveness. This underpins that to raise policy acceptability among the general public one needs to put stronger emphasis on the distributional aspects and potentially other aspects of fairness rather than on policy effectiveness.

Overall, the paper provides evidence that knowledge and information can contribute to improving public acceptance of different types of carbon taxation. Future interventions aimed increasing people's policy acceptance need to improve knowledge about the mechanisms of the policy, including relevant concerns such as fairness and effectiveness.

A potential limitation of the study's survey design is that responses might have been subject to order effects. The information treatment in our study aimed at testing whether perceptions about the effectiveness, fairness and other attributes of a carbon tax would be affected. Therefore, these items were raised to respondents after the information treatment. Responses regarding perceptions of effectiveness, fairness and other attributes were in turn expected to influence acceptability. Hence the order of the survey items is consistent with the issues we wanted to address. Moreover, participants were asked to respond sequentially to different uses of the carbon tax revenues. The initial expression of a perception of, or attitude to, a carbon tax revenue may have primed, anchored or otherwise influenced subsequent responses. However, before starting this sequence, respondents were provided with an initial overview of all five revenue options. This allowed respondents to think about all options before actually responding to them. This feature of our survey design may have limited such effects. Overall, our study arguably complements and supports the findings of other studies using choice or between-subject designs. For example, carbon tax with revenues for climate projects being the most supported option is consistent with the literature, although it does not appear as the first option in the questionnaire. This suggests that order effects may be a negligible concern, but further studies using different designs would be worthwhile to corroborate the present insights.

## Methods

**Data collection.** Our research involves a survey experiment to assess perceptions and attitudes about a carbon tax among the general public, specifically in relation to perceived effectiveness (in terms of reducing $CO_2$ emissions) and fairness of the policy. To test the survey, we undertook a pilot survey among 15 respondents. The data was collected in August 2019 through a web-based questionnaire by the professional survey company "Netquest" (for additional information on the panel, see: http://www.netquest.com/en/panel/sample-calculator/statistical-calculators.html). The sample of citizens was restricted to individuals over 18 years old. The survey was approved by the Committee on Ethics in Animal and Human Experiments of the Autonomous University of Barcelona. Sampling was done by using quotas on age, gender and geographical distribution, making the survey sample representative of the general population on these characteristics. The survey was sent to 3415 Spanish citizens. A total amount of 2534 people accessed the survey, among whom 530 persons were filtered out of the sample, for different reasons, such as answering the control questions incorrectly (11), leaving the survey without completing it (355), or because the quota to which they belonged was already completed (123). This resulted in a final sample size of 2004 and a response rate of 58,68%. The survey company that implemented the questionnaire obtained informed consent from all participants. Respondents took on average about 15 minutes to finish the survey. They were encouraged to participate in it through receipt of a gift voucher.

We divided our sample into two subsamples. The first (group 1, $N = 1004$) received information explaining the functioning of a carbon tax whereas the second sub-sample (group 2, $N = 1000$) did not receive this explanation. Each survey opens with some background information explaining the policy proposal. See Supplementary Notes 2 and 3 for the precise information received by respondents. We did not include a tax rate as our aim was to inquire about peoples' attitudes regarding carbon taxation in general, as a tool or mechanism, independent of a tax rate. People arguably have difficulty translating a concrete tax rate into a personal cost. Moreover, our aim was to obtain general results about how people perceive a carbon tax along with revenue uses, not affected by a specific rate.

The survey was structured into five main sections. In the first, we asked respondents about their self-perceived knowledge about a carbon tax as well as assessed their actual knowledge by providing them several true/false statements about carbon taxation. As there was no scale of carbon-tax knowledge available in the literature, we designed six questions to cover distinct parts of knowledge about such a tax: namely, on tax subject (first question), tax mechanism (second and forth questions), i.e. whether people do not mix it up with standards or permits, tax effects (third and sixth questions), and use of tax money (fifth question). In the second section, we elicited the respondents' beliefs about the effectiveness, fairness and acceptability of a carbon tax and how they consider it may affect them personally (individual effects) and low-income households (distributional effects). Participants could respond on 5-point Likert scales, ranging from "very ineffective/unfair/unacceptable" to "very effective/fair/acceptable" in the case of the former questions and from "I would be much worse off/they would be much worse off" to "I would be much better off/they would be much better off" in the case of the latter two questions. The 5-point scale of acceptability is commonly used in the literature on attitudes to climate policy[36]. Some studies use a dichotomous variable mirroring a ballot decision which particularly makes sense for countries where the political system follows such a procedure. In our research context, Spain, such ballots seldom take place at the national level, hence our choice for a continuous scale. All these questions referred to a carbon tax without specifying or mentioning any details about the tax revenues. In addition, there were questions such as on trust in politicians (see all questions in Supplementary Notes 2 and 3). In the third section, we first introduced participants to the theme of revenues from carbon taxation, and presented them five different uses of the revenues:

(a) Return all the revenues to compensate low-income households.
(b) Support the development of climate projects (e.g. investing in public transport, planting trees, subsidies for renewable energy).
(c) Use half of the revenues to support the development of climate projects and the other half to compensate low-income households.
(d) Return the revenues in equal amount to all households as compensation.
(e) Use half of the revenues to support development of climate projects and the other half to compensate all households in equal amount.

We then asked respondents to express their views about the effectiveness, fairness, personal effects, effects on low-income households and acceptance of a carbon tax for the five different uses of the revenues. In the fourth section, we asked respondents to allocate a percentage of the total carbon tax revenues (100%) to each of the three proposed revenue use options (support the development of climate projects; return the revenues to compensate low-income households; return the revenues in equal amount to all households as compensation). Later, we asked them to indicate which factor—effectiveness or fairness—played a stronger role in this allocation. At the end of the survey, we collected information about socio-demographic and behavioural characteristics of the surveyed individuals (e.g., education, car use, monthly income). Descriptive statistics are presented in Supplementary Table 1.

**Data analysis.** To formally test whether respondents' perceived fairness, effectiveness, personal effects, low-income effects and their acceptability (see Fig. 4) under one revenue use is significantly higher than under an alternative revenue use, we employ a pairwise Mann–Whitney test that explicitly compares responses from the same individuals for stochastic dominance (see Supplementary Fig. 1). We also tested for statistical difference within each revenue use depending on whether information on carbon tax functioning was provided prior to the questions or not using Kruskal–Wallis rank-sum test with Bonferroni correction (see Supplementary Fig. 1).

In order to examine public understanding of a carbon tax, we relied on people's answers to six questions assessing objective knowledge about the functioning of a carbon tax (see Fig. 4). To aggregate our six items, we decided to turn to a Mokken scale analysis, as was used in prior research on citizens' knowledge about climate change[37]. In particular, we follow van der Ark[38] who implemented this method in R. In line with the literature on measuring climate knowledge using the Mokken scale (e.g. Tobler et al.[39]), answers were recoded as dichotomous variables (1 = "correct", 0 = "wrong" and "don't know"), so that we could distinguish people who answered correctly from those who do not. We find only a weak scale for our six questions (Loevinger scalability coefficient of 0.346, which is below the 0.4 threshold to be considered as good). By assessing individual scales for each of the six items (0.334, 0.381, 0.382, 0.294, 0.120, 0.365), we find that the fifth item fits worst into our set of questions. This can be explained by the very low correlation of the positive responses on the fifth statement compared to other five items in our questionnaire. In other words, people who responded correct on this question were unlikely to perform well on other questions, and the other way around. We omitted the respective item, causing the overall scalability coefficient to rise to 0.370 and all individual coefficients above 0.33. All the scalability coefficients are then above 0.3 and hence can be considered according to the literature as weak but still useful[38,40], suggesting that the five items form a reliable one-dimensional scale. Moreover, several other studies on climate knowledge and perceptions obtain similar values of scalability coefficients[39,41].

In order to test how carbon tax acceptability is affected by people's knowledge, information provision about a carbon tax functioning, perceived effectiveness, fairness, effects on low-income households and personal effects, we run ordered logit regressions (Table 1). This choice is supported by the fact that our dependent variables are discrete ordered choices varying from very negative statements (e.g. "very ineffective") to very positive ones ("very effective"). In addition, we run ordered logit regressions with the sample split into six subsamples, namely for each revenue use. The results are consistent with those reported in Table 1, and thus can be seen as robustness test (see Supplementary Table 4). Furthermore, we undertook ordered logit regression to measure the relationship between acceptability and fairness on the one hand, and other carbon tax perceptions on the other (Table 2).

As we have 263 missing values for the variable political orientation and 432 missing values for monthly household income, we ran regressions as shown in Tables 1 and 2 by excluding those missing observations. To test for robustness of our results, we report the same regressions but dropping those two controls and thus using all observations in Supplementary Tables 5 and 6.

To rank predictors of acceptability listed in Table 2 in terms of predictive power we use a gradient boosting machines (GBM) algorithm implemented in the software $R$[42]. GBM is a machine learning algorithm that is based on the principle of classification and regression trees[43]. While basic regression trees suffer from low robustness, GBM avoids this by building a large set (in our case 1000) of simple successive trees with each tree learning and improving on the previous and ensuring robustness of results. To validate the model, we use a 10-fold cross-validation. That means that GBM splits the data ten times randomly into training and testing sets, trains the model on the training data and then evaluates its performance on the testing data. Data analysis has been done in R software (version 4.0.3) and all code used is included in Supplementary Software.

**Reporting summary.** Further information on research design is available in the Nature Research Reporting Summary linked to this article.

## Data availability

All data generated or analysed during this study are included in this published article and its supplementary information. Source data are provided with this paper.

## Code availability

Data analysis has been done in R software. The code is included in Supplementary Software.

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

## Acknowledgements
This research was funded by a RecerCaixa 2016 project titled "Understanding Societal Views on Carbon Pricing" and by an ERC Advanced Grant from the European Research Council (ERC) under the European Union's Horizon 2020 Research and Innovation Programme [grant agreement n° 741087]. I.S. acknowledges financial support from the Russian Science Foundation [RSF grant number 19-18-00262]. We thank the EVOCLIM team for valuable comments.

## Author contributions
S.M.A., S.D., I.S. and J.v.d.B. contributed to the conception and design of the work; I.S. contributed to data analysis; S.M.A., S.D., I.S. and J.v.d.B. contributed to interpretation of data and writing of the manuscript.

## Competing interests
The authors declare no competing interests.
