## [Peer Review File · Nature Communications]

Carbon tax acceptability with information provision and mixed revenue usesREVIEWER COMMENTS

Reviewer #1 (Remarks to the Author):

This paper uses an original survey implemented in Spain over the summer of 2019 to infer, from stated preferences, on public support for carbon taxes. The survey includes a standard informational treatment as well as a novel set of questions aimed at assessing people's understanding of carbon taxes. Later in the survey, the questionnaire also includes a battery of questions on recycling modes, including mixed revenue use.

I am generally sympathetic with the authors' research angle, in particular in what concerns information asymmetries. A recent literature has been considering the role of information asymmetries in hampering public support for cost-effectiveness-enhancing environmental policies, using one of the three following strategies. First, providing people with detailed information on carbon taxes' effects from computable general equilibrium models to correct for information asymmetries, as in Carattini, Kallbekken, and Orlov (2019) or in Carattini et al. (2017), the latter also comparing with a no-information setting. Second, by providing information on the effectiveness of a given instrument in a random fashion, as done for carbon offsets in Baranzini, Borzykowski, and Carattini (2018), for congestion charges in Baranzini, Carattini, and Tesaro (2018), and for carbon taxes, combining stated and revealed preferences, in Carattini et al. (2018). If informational treatments affect people's preferences, then one may infer on the presence of information asymmetries ex treatment. (Note that Douenne and Fabre 2019 provide information at random on distributional effects.) Third, by comparing ex ante and ex post public support and beliefs. If learning through experience leads to belief revision, one may infer on the presence of information asymmetries ex ante. Simple before and after comparisons as in Schuitema, Steg, and Forward (2010); Hansla et al. (2017); Andersson and Nässén (2016) already provided some indications of learning, although the absence of a control group limited the ability of those studies to make causal claims. Cherry, Kallbekken, and Kroll (2014) moved to the lab to enjoy full control, providing evidence for higher acceptability for an efficiency-enhancing Pigouvian policy in treatment arms subject to trial runs. Carattini, Baranzini, and Lalive (2018) exploited a natural experiment to assess belief revisions in presence of an already-treated control group, for which there is no learning involved when the treatment occurs in the treatment group.

Information asymmetries are so important in this recent literature that Carattini, Carvalho, and Fankhauser (2018) include communication strategies as one of the main policy recommendations stemming from the literature on public support for carbon taxes. Following this recommendation, the World Bank commissioned a study on how to best communicate carbon pricing (Marshall et al. 2018).

Here, the authors introduce a new approach, which evaluates respondents' actual knowledge on carbon taxes, their perceived knowledge on carbon taxes, and how these terms correlate with public support for a carbon tax of undefined stringency. They complement this approach with a battery of questions on revenue recycling, which come after the main question on public support, as well as many other questions on carbon tax perceptions.

While I am very sympathetic with the authors' research angle, I have some concerns with the execution of the study, which I outline in what follows. As usual, I start with first-order concerns.

Major concerns

1. Contribution. The conclusion, unfortunately, does not really reflect the contribution of the paper, which in my opinion stems mostly from the fact that it includes an assessment of people's understanding of carbon taxes and of their perceived understanding. As far as the other aspects of the paper are concerned, which seem to appear more prominently in the current version, they are not especially novel. The concluding remarks include the following statements:

- a. "This study reveals that revenue uses can have a marked effect on public acceptability." Obviously, this finding is not novel.
- b. "A carbon tax with all revenues used to support climate projects is the most accepted option".

This result, which precedes information provision, is very much in line with the rest of the literature.

c. "Providing respondents with information about the functioning of the tax can raise acceptability". Not novel either (see above).

d. "The finding that there is a clear preference for using revenues for climate projects turns out to be partly due to respondents insufficiently recognizing that the carbon tax, not its revenue use, is meant to regulate emissions". This is precisely one of the channels studied by the literature on information asymmetries.

A very large amount of papers has been produced in this research area, which implies that there is a risk of reinventing the wheel over and over again (which of course is very bad for science). Following the authors' list of references, I was surprised to discover that Beiser-McGrath and Bernauer (2019) managed to publish (especially well) a paper that provides virtually no new findings to the literature, while still citing the older papers whose findings were replicated. I assume that sometimes peer review can fail. In the case of the current manuscript, there is definitely novelty, in that the authors approach the question of information asymmetry from a new angle, asking people what they (think they) know about carbon taxes. The issue is that, currently, such novelty, and lack thereof in other areas, is not reflected in the paper.

2. Context. The paper uses data for Spain. According to the Appendix summary of the literature in Carattini, Carvalho, and Fankhauser (2018), only the lab study by Heres, Kallbekken, and Galarraga (2017) used a Spanish sample. The current manuscript uses a sample for a broader population than just students in Bilbao. While whether the sample is representative or not cannot be determined based on the information currently provided, the focus on the general public of Spain adds to the paper's contribution, in my opinion. In this respect, it is surprising that the authors do not say a word about the context that they study. While strict word limits apply to manuscripts submitted to this Journal, to the best of my knowledge there are no word limits to the Supplementary Information, which would allow the authors to shuffle content around (more on this below). Hence, I would consider a short section on the context that the authors study of use. According to World Bank (2019), Spain has a carbon tax since 2014. I assume that, to justify the focus of the paper, either the tax's coverage is limited, or the tax rate is relatively low (\$17 per ton of CO₂ according to the same source). Providing information about the context that the authors analyze is crucial. Whether people support a carbon tax in a context where there is one already is different from the context whether there is none. Whether people know how carbon taxes work in a context where there is one already is different from the context whether there is none. In Switzerland, for instance, very few individuals know about the carbon tax, and even fewer know how revenues are redistributed (INFRAS 2015; Baranzini and Carattini 2017; Carattini et al. 2017). Hence the recommendation in Carattini, Carvalho, and Fankhauser (2018) to also have ex-post communication strategies (besides ex-ante), since carbon pricing is not as visible as congestion charges or pricing garbage by the bag.

3. Survey design. The survey is designed in a very unconventional way.

a. The questionnaire starts with a series of questions on understanding of carbon taxes. Then, part of the sample is informed about how carbon taxes work. One would assume that the treatment occurs randomly, but that unfortunately does not appear explicitly in the paper. Balance of covariates for treatment and control are not provided, even though the authors have quite a few socioeconomic characteristics to compare. No data were provided, to the best of my knowledge, with the submission.

b. Even before asking about public support, the questionnaire raises questions about effectiveness, fairness, personal impacts, distributional effects, and the burden of the tax on producers versus consumers (i.e. the relationship between the price elasticities of supply and demand). The authors even ask to what extent they believe that politicians should be trusted in implementing the carbon tax. These questions may already introduce some substantial priming in the respondent. Only after this rather non-neutral battery of questions, respondents are asked about public support, rather than the other way around. The question about public support is asked in a continuous way, at odds with most of the literature (if there is a specific rationale, it does not appear explicitly in the paper). Note that at this point the respondent does not know anything about how revenues would be used in this hypothetical scenario. Most strikingly, the authors do not define a carbon tax rate,

in contrast with the norm in the literature. Hence, the question is very similar, in its vagueness, to what the European Social Survey uses. It is a bit like asking whether one is in favor of income taxes, without specifying what are the rates and how money will be used. I can picture myself trying to interpret the authors' questions. The authors use the term "unspecified revenue use" for something that may indicate funding the general budget for some respondents (usually pushing public support down) or the preferred revenue use for some respondents. I find priming and vagueness very odd in general and even more so for a study that acknowledges and focuses on the importance of design and communication.

c. Then the questionnaire enters a second part, in which revenue use is introduced and additional questions about public support are asked. Most researchers would have addressed the authors' research question with a split sample design or a discrete choice experiment, to prevent priming respondents and boosting pro-social bias. The more the questionnaire advances, the less credible the answers become.

d. In the second part of the questionnaire, the authors refer to the revenues that a €5 per ton of CO₂ carbon tax would raise in Spain (about €1.3 billion). However, it is still not obvious whether the respondent is "voting" on a €5 per ton of CO₂ carbon tax (lower than the current tax rate).

e. Finally, the authors introduce mixed revenue uses. Mixed revenue uses arrive so late in the questionnaire that it is not obvious what can be done with those questions. A shorter survey with a larger pool and a split sample design would have allowed having many more (credible) findings. Furthermore, it is not obvious why respondents can mix revenues to compensate both low-income households and revenues to compensate all households, especially when returning the revenues to all households already makes the reform progressive.

4. Sample. The authors claim that the survey is representative of the underlying population. However, there is no evidence in support of this claim. Hence, my understanding is that what the authors meant is that the survey was conceived by a marketing company, through stratification, to be representative of the underlying population. Whether it actually is, is an empirical question, especially given that respondents were paid to participate in the study. Hence, Table B.1 should also include a column for the underlying population. Further, as mentioned above, another table should discuss the balance of covariates between control and treatment groups. Finally, the authors do not provide one of the most important pieces of information when it comes to surveys, which is the response rate.

5. Assessment of people's knowledge. One very curious finding in the paper is that "assessed knowledge" exceeds "self-perceived knowledge". This finding is not obvious to explain. The analogy with genetically modified food obviously does not work, since it refers to exactly the opposite case: "This result is in line with the findings of a prior study that for controversial topics such as genetically modified food, and to a lesser extent climate change, extreme opponents have limited understanding of the matter but nevertheless think they know the most". In this context, with assessed knowledge higher than self-perceived knowledge, there are two possible explanations (or two come to my mind): (1) either the questions that the authors ask are too generic and do not capture the lack of knowledge from which respondents really suffer; or (2) respondents underestimate their knowledge level on carbon taxes. It would be useful if the authors could shed light on this question, assuming that their questionnaire allows them to do so.

6. Main results. There are a few results that can be interesting, despite the way the survey was designed.

a. Figure 3 presents the relationship between assessed knowledge and acceptability with and without information provision. Hence, the goal is to estimate heterogeneous treatment effects (information provision being the treatment) along the assessed knowledge dimension. I would still keep Figure 3, but a standard regression with heterogeneous treatment effects (interaction term) and control variables would be much more transparent and easier to interpret rather than eyeballing potentially significant differences by comparing coefficients and their confidence intervals. A quadratic term may be also introduced.

b. The effect of information provision on acceptability seems noisy once considering different levels of assessed knowledge (Figure 3), but is likely going to be significant once considering the entire sample, as standard errors decrease in size. Control variables can also reduce noise. I would start from that finding (average treatment effects). Then, one can assess heterogeneous treatment effects.

c. When the authors consider belief revision, for instance on fairness or effectiveness, as a result of information provision, it would be useful to compare with the belief revision occurring with learning from experience (Carattini, Baranzini, and Lalive 2018).

7. Additional figures and tables.

a. The authors find that “The two mixed revenue uses increase acceptability compared to unspecified revenue use, but not compared to spending all revenues on supporting climate projects.” That makes sense if people assumed that the unspecified revenue use would have meant a transfer to the general budget. Unfortunately, we cannot tell what respondents thought when answering the survey.

b. It is interesting that respondents find very small effects on preferences for the use of revenues following information provision. That may be due to the very generic informational treatment, although all the priming that occurred before acceptability and revenue use were introduced might also have muddied the waters.

c. The authors analyze the effect of information provision on acceptability conditional on the use of revenues. However, both acceptability and use of revenues are endogenous variables. Hence, I do not think that the authors could make statements such as “Third, we find that information provision tends to result in more favourable perceptions of acceptability, fairness and effectiveness of a carbon tax for unspecified revenues and to a lesser extent for revenues going in equal amount to poor households and climate projects, while it shows no significant effect for the other revenue schemes.”

d. When analyzing how respondents would allocate tax revenues to different uses if they had complete freedom to decide about this, I would compare with the equivalent question in Carattini et al. (2017).

e. Pages 8 and 9 provide results with endogenous variables on both the left-hand and right-hand side, which I do not consider as they should not appear in the paper. A shorter, well-designed questionnaire can do much more than a long questionnaire in which everything becomes endogenous and moves from explanatory to explained variable depending on the estimation (which of course is not appropriate).

8. Robustness tests. The authors mention a single robustness test, which, however, is only “available upon request”. The authors’ approach does not seem very transparent.

9. Imputing missing values. I am very concerned about imputing variables such as political orientation and income based on other covariates. The best course of action is to drop missing observations and to present those results, along potentially with a column without control variables (and a larger sample size), for comparison. If the results “do not alter significantly”, then there should be no issue.

Minor concerns

1. I would encourage the authors to share the questionnaire in Spanish too. It is possible that some authors just want to use your set of questions. We often do not reinvent the wheel when designing surveys and use questions from the World Values Surveys or the European Social Survey. Some surveys clearly undergo much more thinking when designed, but I cannot exclude that someone would use this one too for some purposes.

2. References sometimes mix original studies and review papers, even if the review papers refer to the same original study. It may help the reader to refer to the original study for the finding and to the review paper for whatever additional intuition it provides, if any.

3. It is not clear whether you designed 15 pilot questionnaires – then it would be very surprising that you used the current one – or whether you piloted your questionnaire on 15 respondents.

4. The description of the Mokken scale in the main body of text and the data section is very unclear. It seems that your variables do not reach the minimum threshold of validity to be used. Then, you should not use them and switch to a different approach.

5. Sometimes the authors make comparisons without clarifying what the baseline case is. For instance: “Respondents considered returning revenues to poor households and the combination of this with supporting climate projects as the least effective in terms of emissions reductions.” Compared to what?

References

- Andersson, David, and Jonas Nässén. 2016. "The Gothenburg Congestion Charge Scheme: A Pre-Post Analysis of Commuting Behavior and Travel Satisfaction." *Journal of Transport Geography* 52: 82–89.
- Baranzini, Andrea, Nicolas Borzykowski, and Stefano Carattini. 2018. "Carbon Offsets out of the Woods? Acceptability of Domestic vs. International Reforestation Programmes in the Lab." *Journal of Forest Economics* 32: 1–12.
- Baranzini, Andrea, and Stefano Carattini. 2017. "Effectiveness, Earmarking and Labeling: Testing the Acceptability of Carbon Taxes with Survey Data." *Environmental Economics and Policy Studies* 19 (1): 197–227.
- Baranzini, Andrea, Stefano Carattini, and Linda Tesauro. 2018. "The Geneva Congestion Charge: Rationale, Design, and Acceptability." *World Congress of Environmental and Resource Economists*.
- Beiser-McGrath, Liam F., and Thomas Bernauer. 2019. "Could Revenue Recycling Make Effective Carbon Taxation Politically Feasible?" *Science Advances* 5 (9): eaax3323.
- Carattini, Stefano, Andrea Baranzini, and Rafael Lalive. 2018. "Is Taxing Waste a Waste of Time? Evidence from a Supreme Court Decision." *Ecological Economics* 148: 131–51.
- Carattini, Stefano, Andrea Baranzini, Philippe Thalmann, Frédéric Varone, and Frank Vöhringer. 2017. "Green Taxes in a Post-Paris World: Are Millions of Nays Inevitable?" *Environmental and Resource Economics* 68 (1): 97–128.
- Carattini, Stefano, Maria Carvalho, and Sam Fankhauser. 2018. "Overcoming Public Resistance to Carbon Taxes." *Wiley Interdisciplinary Reviews: Climate Change* 9 (5): e531.
- Carattini, Stefano, Anomitro Chatterjee, Todd L. Cherry, and Steffen Kallbekken. 2018. "Climate Levy, Informational Messages, Acceptability, and Treatment Effects of Framing Effectiveness and Earmarking (CLIMATEFEE)." AEA RCT Registry. <https://doi.org/10.1257/rct.3476-1.0>.
- Carattini, Stefano, Steffen Kallbekken, and Anton Orlov. 2019. "How to Win Public Support for a Global Carbon Tax." *Nature* 565 (7739): 289–91.
- Cherry, Todd L., Steffen Kallbekken, and Stephan Kroll. 2014. "The Impact of Trial Runs on the Acceptability of Environmental Taxes: Experimental Evidence." *Resource and Energy Economics* 38 (C): 84–95.
- Douenne, Thomas, and Adrien Fabre. 2019. "Can We Reconcile French People with the Carbon Tax? Disentangling Beliefs from Preferences." 2019.10. Working Papers. FAERE - French Association of Environmental and Resource Economists. <https://ideas.repec.org/p/fae/wpaper/2019.10.html>.
- Hansla, André, Erik Hysing, Andreas Nilsson, and Johan Martinsson. 2017. "Explaining Voting Behavior in the Gothenburg Congestion Tax Referendum." *Transport Policy* 53: 98–106.
- Heres, David R., Steffen Kallbekken, and Ibon Galarraga. 2017. "The Role of Budgetary Information in the Preference for Externality-Correcting Subsidies over Taxes: A Lab Experiment on Public Support." *Environmental and Resource Economics* 66 (1): 1–15.
- INFRAS. 2015. "Klimaschutz Und Grüne Wirtschaft - Was Meint Die Bevölkerung? Ergebnisse Einer Repräsentativen Bevölkerungsbefragung - Schlussbericht." Zurich: Bundesamt für Umwelt.
- Marshall, George, Darragh Conway, Robin Webster, Louise Comeau, Daniel James Besley, and Isabel Saldarriaga Arango. 2018. "Guide to Communicating Carbon Pricing." 132534. The World Bank. <http://documents.worldbank.org/curated/en/668481543351717355/Guide-to-Communicating-Carbon-Pricing>.
- Schuitema, Geertje, Linda Steg, and Sonja Forward. 2010. "Explaining Differences in Acceptability before and Acceptance after the Implementation of a Congestion Charge in Stockholm." *Transportation Research Part A: Policy and Practice* 44 (2): 99–109.
- World Bank. 2019. "State and Trends of Carbon Pricing – 2019." Washington, DC.

Reviewer #2 (Remarks to the Author):

This is a worthwhile topic that will be of interest to others. The methodology seems fine for the most part, and the sample is good. I think the paper is likely to be acceptable. But there are a few areas where a lack of clarity should be addressed. Especially, it is difficult to follow some of the reported results.

Even the abstract could do a better job of summarizing the important information.

One of the primary manipulations is presenting half the participants with information about the carbon tax. Did providing information actually increase either perceived or assessed knowledge? This would be worth assessing.

The second main independent variable is the use of the revenue from the carbon tax. It was not clear to me until I read the more detailed Methods that fairness and acceptability were rated both in general and for specific revenue uses. Are the general or specific ratings used in the data analyses? This is not clear.

Individual fairness and individual wellbeing are not the same thing. They should not be conflated (e.g. line 79). Similarly, reducing the wellbeing of those who are worst off is not the same as increasing distributional fairness, though they are clearly related. In addition, describing these measures as measures of fairness makes it more confusing that you also measured what appear to be straightforward ratings of fairness. You should clarify this.

The figures are good.

I'm also confused by Table 2. The description refers to the third column twice, and I think the second time the fourth column is meant. But I'm not sure what the different analyses are that are being presented in the third and fourth column.

I spent some time going back and forth between the description of results and the description of the methods to try to better understand the analyses, but the reader should not have to work this hard. Please try to clarify the description of the results.

Responses to reviewers

Manuscript ID NCOMMS-20-13391-T entitled "Carbon tax acceptability with information provision and mixed revenue uses"

We are grateful to the two reviewers for their detailed and constructive comments. Below we explain how we addressed them (our response is marked in italics).

Response to reviewer #1

This paper uses an original survey implemented in Spain over the summer of 2019 to infer, from stated preferences, on public support for carbon taxes. The survey includes a standard informational treatment as well as a novel set of questions aimed at assessing people's understanding of carbon taxes. Later in the survey, the questionnaire also includes a battery of questions on recycling modes, including mixed revenue use.

I am generally sympathetic with the authors' research angle, in particular in what concerns information asymmetries. A recent literature has been considering the role of information asymmetries in hampering public support for cost-effectiveness-enhancing environmental policies, using one of the three following strategies. First, providing people with detailed information on carbon taxes' effects from computable general equilibrium models to correct for information asymmetries, as in Carattini, Kallbekken, and Orlov (2019) or in Carattini et al. (2017), the latter also comparing with a no-information setting. Second, by providing information on the effectiveness of a given instrument in a random fashion, as done for carbon offsets in Baranzini, Borzykowski, and Carattini (2018), for congestion charges in Baranzini, Carattini, and Tesauro (2018), and for carbon taxes, combining stated and revealed preferences, in Carattini et al. (2018). If informational treatments affect people's preferences, then one may infer on the presence of information asymmetries ex treatment. (Note that Douenne and Fabre 2019 provide information at random on distributional effects.) Third, by comparing ex ante and ex post public support and beliefs. If learning through experience leads to belief revision, one may infer on the presence of information asymmetries ex ante. Simple before and after comparisons as in Schuitema, Steg, and Forward (2010); Hansla et al. (2017); Andersson and Nässén (2016) already provided some indications of learning, although the absence of a control group limited the ability of those studies to make causal claims. Cherry, Kallbekken, and Kroll (2014) moved to the lab to enjoy full control, providing evidence for higher acceptability for an efficiency-enhancing Pigouvian policy in treatment arms subject to trial runs. Carattini, Baranzini, and Lalive (2018) exploited a natural experiment to assess belief revisions in presence of an already-treated control group, for which there is no learning involved when the treatment occurs in the treatment group.

Information asymmetries are so important in this recent literature that Carattini, Carvalho, and Fankhauser (2018) include communication strategies as one of the main policy recommendations stemming from the literature on public support for carbon taxes. Following

this recommendation, the World Bank commissioned a study on how to best communicate carbon pricing (Marshall et al. 2018).

Here, the authors introduce a new approach, which evaluates respondents' actual knowledge on carbon taxes, their perceived knowledge on carbon taxes, and how these terms correlate with public support for a carbon tax of undefined stringency. They complement this approach with a battery of questions on revenue recycling, which come after the main question on public support, as well as many other questions on carbon tax perceptions.

Thank you for sketching this literature context and acknowledging the innovation and relevance of our contribution. We already referred in our paper to some of the mentioned studies, in particular the literature review by Carattini, Carvalho, and Fankhauser (2018). We have added some of the above studies that were not yet mentioned, except the ones that did not specifically address carbon taxation. For more details, see specific responses below.

While I am very sympathetic with the authors' research angle, I have some concerns with the execution of the study, which I outline in what follows. As usual, I start with first-order concerns.

Major concerns

1. Contribution. The conclusion, unfortunately, does not really reflect the contribution of the paper, which in my opinion stems mostly from the fact that it includes an assessment of people's understanding of carbon taxes and of their perceived understanding. As far as the other aspects of the paper are concerned, which seem to appear more prominently in the current version, they are not especially novel. The concluding remarks include the following statements:

a. "This study reveals that revenue uses can have a marked effect on public acceptability." Obviously, this finding is not novel.

b. "A carbon tax with all revenues used to support climate projects is the most accepted option". This result, which precedes information provision, is very much in line with the rest of the literature.

c. "Providing respondents with information about the functioning of the tax can raise acceptability". Not novel either (see above).

d. "The finding that there is a clear preference for using revenues for climate projects turns out to be partly due to respondents insufficiently recognizing that the carbon tax, not its revenue use, is meant to regulate emissions". This is precisely one of the channels studied by the literature on information asymmetries.

Thank you. Your remarks have motivated us to better highlight the contributions of our study. First, single revenue uses have indeed received considerable attention in previous studies. But to our knowledge, it remains unclear how acceptability, effectiveness and other perceptions are associated with mixed revenue uses, and how these compare with single uses. According to the

earlier mentioned review by Carattini, Carvalho, and Fankhauser (2018) “hybrid strategies mixing different revenue recycling options” are “underexplored in the literature”. Second, some evidence shows no effects of information provision, as we discussed in the original paper. Moreover, the reviewer cites some references to studies on information provision (Douenne and Fabre, 2019; Carattini et al., 2018) which appear to be working papers. We and Nature prefer to focus on evidence published in journals. Overall, we think the evidence for this issue is still relatively thin. In addition, in view of debates concerning the replication crisis in some social sciences, we consider a cumulative evidence base as something worth striving for. Anyway, we have adapted the text now to better clarify what are the main innovative findings and which findings are in line with earlier studies.

A very large amount of papers has been produced in this research area, which implies that there is a risk of reinventing the wheel over and over again (which of course is very bad for science). Following the authors' list of references, I was surprised to discover that Beiser-McGrath and Bernauer (2019) managed to publish (especially well) a paper that provides virtually no new findings to the literature, while still citing the older papers whose findings were replicated. I assume that sometimes peer review can fail. In the case of the current manuscript, there is definitely novelty, in that the authors approach the question of information asymmetry from a new angle, asking people what they (think they) know about carbon taxes. The issue is that, currently, such novelty, and lack thereof in other areas, is not reflected in the paper.

Thank you for this comment. We have adapted the abstract, introduction and conclusions to better highlight one of the novelties of our paper that the reviewer is pointing out. We have further added a paragraph to the introduction on studies addressing information asymmetries to contextualize our contribution. Hopefully, the reviewer agrees that replicating certain results for different country contexts and under different framings (e.g. comparing between different revenue use options and between information provision or not) also counts as a valuable scientific contribution.

2. Context. The paper uses data for Spain. According to the Appendix summary of the literature in Carattini, Carvalho, and Fankhauser (2018), only the lab study by Heres, Kallbekken, and Galarraga (2017) used a Spanish sample. The current manuscript uses a sample for a broader population than just students in Bilbao. While whether the sample is representative or not cannot be determined based on the information currently provided, the focus on the general public of Spain adds to the paper's contribution, in my opinion. In this respect, it is surprising that the authors do not say a word about the context that they study. While strict word limits apply to manuscripts submitted to this Journal, to the best of my knowledge there are no word limits to the Supplementary Information, which would allow the authors to shuffle content around (more on this below). Hence, I would consider a short section on the context that the authors study of use. According to World Bank (2019), Spain has a carbon tax since 2014. I assume that, to justify the focus of the paper, either the tax's coverage is limited, or the tax rate is relatively low (\$17 per ton of CO₂ according to the same source). Providing information about the context that the authors analyze is crucial. Whether

people support a carbon tax in a context where there is one already is different from the context whether there is none. Whether people know how carbon taxes work in a context where there is one already is different from the context whether there is none. In Switzerland, for instance, very few individuals know about the carbon tax, and even fewer know how revenues are redistributed (INFRAS 2015; Baranzini and Carattini 2017; Carattini et al. 2017). Hence the recommendation in Carattini, Carvalho, and Fankhauser (2018) to also have ex-post communication strategies (besides ex-ante), since carbon pricing is not as visible as congestion charges or pricing garbage by the bag.

Thank you for this suggestion. We added explanations to the main text and in Appendix A that Spain has never had what is commonly considered a carbon tax. Instead, the World Bank interprets a tax aimed at reducing fluorinated greenhouse gases (F-gases) as a carbon tax (see the entry for Spain in the Carbon Pricing Dashboard of The World Bank: https://carbonpricingdashboard.worldbank.org/map_data). Such a tax can of course be interpreted as an implicit carbon price, but to call it an explicit carbon tax would be incorrect. Of course, Spain has a carbon price due to being part of the EU-ETS, but this is not a tax. For this reason refereed reviews of carbon taxation do not to mention Spanish (e.g., Haites, 2018).

3. Survey design. The survey is designed in a very unconventional way.

a. The questionnaire starts with a series of questions on understanding of carbon taxes. Then, part of the sample is informed about how carbon taxes work. One would assume that the treatment occurs randomly, but that unfortunately does not appear explicitly in the paper. Balance of covariates for treatment and control are not provided, even though the authors have quite a few socioeconomic characteristics to compare. No data were provided, to the best of my knowledge, with the submission.

We use quotas on age, gender and geographical distribution to ensure that treatment and control groups have a similar distribution regarding these three covariates. We now explicitly mention this in the manuscript. Furthermore, we compared the two subsamples using the Kruskal-Wallis rank sum test with Bonferroni correction taking into account other covariates, such as their climate concern, and they are not statistically different (p values always above 0.1). We have added this information now in a table in Appendix C.

b. Even before asking about public support, the questionnaire raises questions about effectiveness, fairness, personal impacts, distributional effects, and the burden of the tax on producers versus consumers (i.e. the relationship between the price elasticities of supply and demand). The authors even ask to what extent they believe that politicians should be trusted in implementing the carbon tax. These questions may already introduce some substantial priming in the respondent. Only after this rather non-neutral battery of questions, respondents are asked about public support, rather than the other way around. The question about public support is asked in a continuous way, at odds with most of the literature (if there is a specific rationale, it does not appear explicitly in the paper). Note that at this point the respondent does not know anything about how revenues would be used in this hypothetical scenario. Most strikingly, the authors do not define a carbon tax rate, in

contrast with the norm in the literature. Hence, the question is very similar, in its vagueness, to what the European Social Survey uses. It is a bit like asking whether one is in favor of income taxes, without specifying what are the rates and how money will be used. I can picture myself trying to interpret the authors' questions. The authors use the term "unspecified revenue use" for something that may indicate funding the general budget for some respondents (usually pushing public support down) or the preferred revenue use for some respondents. I find priming and vagueness very odd in general and even more so for a study that acknowledges and focuses on the importance of design and communication.

Regarding priming, we believe that it cannot be entirely excluded in any questionnaire as one question is always preceded by another that might create a priming effect (except the first question of course). The information treatment in our study aimed at testing whether perceptions about the effectiveness of a carbon tax would be affected. Therefore, this item was the first one raised to respondents after the information treatment. Perceptions of effectiveness were in turn expected to influence acceptability. Hence the order of the survey items is consistent with the issues we wanted to address. Nevertheless, we agree that some question order effects might be at work which might warrant a specialized research study.

The reviewer suggests that how we measure tax acceptability is "at odds with most of the literature". We think this depends on what one considers as "the literature". It is true that choice experiments measure support in a dichotomous way. However, other types of studies use continuous scales. Just one example of this is the study by Schuitema, Steg, and Forward (2010) which the reviewer cited above. Other examples can be found in the literature reviews by Carattini et al. (2018) and Maestre-Andrés et al. (2019).

The absence of a tax rate is explained by our aim to inquire about people's attitudes regarding carbon taxation in general, as a tool or mechanism to achieve CO₂ emissions reduction. This is rather independent of tax rate. Our aim was to obtain results not affected by a specific rate. One can anyway doubt if laypersons are well able to judge the specific implications of a tax rate. People will have difficulty to translate a concrete tax rate into an actual cost for their household. Another reason is that any rate would be arbitrary, so that extensive sensitivity analysis would be needed, leading to an entirely different focus of the study. Moreover, it is widely agreed that starting with a low rate which is then gradually increased is a good approach to moderate negative impacts on the economy, as all agents have time to anticipate and prepare responses, such as costly investments. To capture this would obviously complicate the study tremendously. Nevertheless, we acknowledge this point now in the manuscript and motivate our decision to do not include a tax rate.

c. Then the questionnaire enters a second part, in which revenue use is introduced and additional questions about public support are asked. Most researchers would have addressed the authors' research question with a split sample design or a discrete choice experiment, to prevent priming respondents and boosting pro-social bias. The more the questionnaire advances, the less credible the answers become.

We split the sample for the information provision but not for the revenue uses because we intentionally informed respondents about all five revenue use options before asking the questions covering each revenue use, allowing people to compare them and carefully express their perceptions and preferences. This initial overview of all revenue options limits priming effects. Hence, we feel our design is adequate and complements the findings of other studies using choice or between-subject designs. Note also that we find a carbon tax with revenues for climate projects is the most supported option, although it does not appear as the first option in the questionnaire. This preference for climate/environmental earmarking is consistent with the literature, as the reviewer already indicated, and suggests that priming is not a serious concern.

d. In the second part of the questionnaire, the authors refer to the revenues that a €5 per ton of CO₂ carbon tax would raise in Spain (about €1.3 billion). However, it is still not obvious whether the respondent is “voting” on a €5 per ton of CO₂ carbon tax (lower than the current tax rate).

This was an illustrative example to enable the respondent to imagine a quantity of revenue collected through the tax. We now have clarified this better in the manuscript.

e. Finally, the authors introduce mixed revenue uses. Mixed revenue uses arrive so late in the questionnaire that it is not obvious what can be done with those questions. A shorter survey with a larger pool and a split sample design would have allowed having many more (credible) findings. Furthermore, it is not obvious why respondents can mix revenues to compensate both low-income households and revenues to compensate all households, especially when returning the revenues to all households already makes the reform progressive.

We already noted in our initial manuscript that “we presented participants with five different uses of the revenues” before they were asked to express their views on the distinct schemes. Hence, mixed revenue uses appear simultaneously with single revenue uses in the questionnaire. Perhaps the reviewer misunderstood the structure of the questionnaire. We have checked the relevant text and slightly modified it to improve clarity.

4. Sample. The authors claim that the survey is representative of the underlying population. However, there is no evidence in support of this claim. Hence, my understanding is that what the authors meant is that the survey was conceived by a marketing company, through stratification, to be representative of the underlying population. Whether it actually is, is an empirical question, especially given that respondents were paid to participate in the study. Hence, Table B.1 should also include a column for the underlying population. Further, as mentioned above, another table should discuss the balance of covariates between control and treatment groups. Finally, the authors do not provide one of the most important pieces of information when it comes to surveys, which is the response rate.

We have added a column in the table (now C.1 in Appendix C) with the underlying population statistics in order to show the representativeness of our sample in terms of age, gender and

political orientation. In a previous comment we already addressed the representativeness of the control and treatment groups. We have now also specified the response rate in the manuscript, as well as further related indicators.

5. Assessment of people's knowledge. One very curious finding in the paper is that "assessed knowledge" exceeds "self-perceived knowledge". This finding is not obvious to explain. The analogy with genetically modified food obviously does not work, since it refers to exactly the opposite case: "This result is in line with the findings of a prior study that for controversial topics such as genetically modified food, and to a lesser extent climate change, extreme opponents have limited understanding of the matter but nevertheless think they know the most". In this context, with assessed knowledge higher than self-perceived knowledge, there are two possible explanations (or two come to my mind): (1) either the questions that the authors ask are too generic and do not capture the lack of knowledge from which respondents really suffer; or (2) respondents underestimate their knowledge level on carbon taxes. It would be useful if the authors could shed light on this question, assuming that their questionnaire allows them to do so.

The statements to elicit actual knowledge of people about a carbon tax are kept rather simple so that a layperson can answer them. Indeed, many people have correct responses to multiple statements. On the other hand, they are not too simple, indicated by few people responding correctly to 4 or all 5 statements. See Figure 1b for more details. Hence, we think the questions strike a good balance between being neither too easy nor too difficult.

Your second suggestion, that respondents underestimate their knowledge on carbon taxation, is indeed what we find for people accepting the carbon tax (Figure 2). This is somewhat consistent with other research on confidence in knowledge about climate change (Fischer et al., 2019). We have now discussed this possible explanation in the manuscript.

However, the analogy with genetically modified food (GMF) is correct for people with low acceptability, not the whole sample. As we say in the paper, and the referee may have overlooked, "people with a low acceptability perceive themselves as having much knowledge about carbon taxation, i.e. when we compare them with the rest of the sample. Even so, the measure of assessed knowledge does not support this perception". And this is the situation reported on GMF by Fernback et al. (2019).

6. Main results. There are a few results that can be interesting, despite the way the survey was designed.

a. Figure 3 presents the relationship between assessed knowledge and acceptability with and without information provision. Hence, the goal is to estimate heterogeneous treatment effects (information provision being the treatment) along the assessed knowledge dimension. I would still keep Figure 3, but a standard regression with heterogeneous treatment effects (interaction term) and control variables would be much more transparent and easier to interpret rather than eyeballing potentially significant differences by comparing coefficients and their confidence intervals. A quadratic term may be also introduced.

We have followed the reviewer's advice and have undertaken an ordered logit regression for unspecified revenue use only (just like in Figure 3), with acceptability as dependent variable and controlling for basic characteristics of respondents (Table C.3 in Appendix C). However, we have the impression that the referee has overlooked that in Figure 3 we not only report means and confidence intervals but also present p-values of a statistical test. We write this in the following sentences: "Our results show that people receiving information about the functioning of a carbon tax tend to have a higher probability to accept it, particularly those with already relatively high assessed knowledge, i.e. with levels 3 and 4. The difference between the subsamples is significant at 5 and 10% levels, respectively, according to a Mann-Whitney test (p values at the bottom of Figure 3)." In fact, running this test instead of a regression model with controls is a simpler way to make the same point and in fact is easier to read. What the referee suggests to do with an interaction term and control variables is what we do later in Table 1 for different revenue uses.

b. The effect of information provision on acceptability seems noisy once considering different levels of assessed knowledge (Figure 3), but is likely going to be significant once considering the entire sample, as standard errors decrease in size. Control variables can also reduce noise. I would start from that finding (average treatment effects). Then, one can assess heterogeneous treatment effects.

Table 1 reports the average treatment effect, i.e. the impact of information provision on acceptability for different revenue uses. As you can see, the effect of information provision on acceptability is positive and significant for some but not all revenue uses. We stress this finding now more in the paper.

c. When the authors consider belief revision, for instance on fairness or effectiveness, as a result of information provision, it would be useful to compare with the belief revision occurring with learning from experience (Carattini, Baranzini, and Lalive 2018).

We think it is theoretically and empirically risky to make this comparison, as they are two fairly different things. Individuals changing their beliefs after learning from experience is a clearly dynamic process. But our study does not compare individual's beliefs over time, only averages between two groups of respondents. Hence, we cannot qualify this as "learning" or "belief revision". By the way, the paper you refer to is about another application anyway, namely a garbage tax where learning is perhaps more relevant and empirically observed.

7. Additional figures and tables.

a. The authors find that "The two mixed revenue uses increase acceptability compared to unspecified revenue use, but not compared to spending all revenues on supporting climate projects." That makes sense if people assumed that the unspecified revenue use would have meant a transfer to the general budget. Unfortunately, we cannot tell what respondents thought when answering the survey.

Our objective is to examine whether making specific revenue uses salient increases people's acceptability compared to no specification of the revenues. This does not deny that respondents can have different thoughts about the unspecified revenue use option. We simply find a significant average effect, which is relevant. Note that "unspecified" revenue use means communicating a carbon tax without raising the issue of revenues. Many public discussions about carbon taxes have this character, i.e. do not dwell on the revenues, but rather on other aspects. This underpins the relevance of including such a case in our study design.

b. It is interesting that respondents find very small effects on preferences for the use of revenues following information provision. That may be due to the very generic informational treatment, although all the priming that occurred before acceptability and revenue use were introduced might also have muddied the waters.

Please bear in mind that the information treatment does not explicitly refer to use of the revenues, but to the functioning of the policy. Our results clearly show there is an increase in acceptability (and also in perceived effectiveness and fairness) for the carbon tax which does not specify revenue use, and to a lesser extent for one of the mixed revenue use options.

c. The authors analyze the effect of information provision on acceptability conditional on the use of revenues. However, both acceptability and use of revenues are endogenous variables. Hence, I do not think that the authors could make statements such as "Third, we find that information provision tends to result in more favourable perceptions of acceptability, fairness and effectiveness of a carbon tax for unspecified revenues and to a lesser extent for revenues going in equal amount to poor households and climate projects, while it shows no significant effect for the other revenue schemes."

Sorry, but we do not understand this remark. Perhaps there is a misunderstanding here. The revenue use scenario is an independent variable that we use to explain acceptability. For each revenue use we ask respondents how acceptable they find it and how they perceive its effectiveness and fairness. We pool the six scenarios (12024 observations in total) and run the regressions controlling for the relevant revenue-use scenario.

d. When analyzing how respondents would allocate tax revenues to different uses if they had complete freedom to decide about this, I would compare with the equivalent question in Carattini et al. (2017).

We followed the reviewer's advice and now compare our results with Carattini et al. (2017).

e. Pages 8 and 9 provide results with endogenous variables on both the left-hand and right-hand side, which I do not consider as they should not appear in the paper. A shorter, well-designed questionnaire can do much more than a long questionnaire in which everything becomes endogenous and moves from explanatory to explained variable depending on the estimation (which of course is not appropriate).

As we explained above, the explanatory variables in Table 1 are all exogenous. So there is no problem in interpreting them the way we do. Concerning Table 2, it is true that we have endogenous variables on the right hand side (perception of effectiveness, fairness, and personal and low-income effects). But when interpreting the results, we abstain from stating any causation statement (we use word like “relation” and “predictor”), and concentrate on correlations. Our main purpose in this part of the analysis is to identify whether effectiveness or fairness is a better predictor of acceptability. Therefore, we think our conclusions hold. Note further that we now added results from applying a gradient boosting machines algorithm, which further support our conclusion that fairness perception is the best predictor of acceptability.

8. Robustness tests. The authors mention a single robustness test, which, however, is only “available upon request”. The authors’ approach does not seem very transparent.

We agree with you and now report the robustness tests in Appendix C, Tables C.3, C.4, C.5 and C.6.

9. Imputing missing values. I am very concerned about imputing variables such as political orientation and income based on other covariates. The best course of action is to drop missing observations and to present those results, along potentially with a column without control variables (and a larger sample size), for comparison. If the results “do not alter significantly”, then there should be no issue.

As you suggest, we repeated the analysis for the sample where missing observations are dropped. The results show little differences except for a somewhat lower significance of information provision for mixed revenue use (PoorHH&Climate). We replaced now our previous results with these ones, while reporting in Tables C5 and C6 in Appendix C results for all 12024 observations, dropping the two controls of monthly income and political orientation (as these contain in total 3498 missing observations).

Minor concerns

1. I would encourage the authors to share the questionnaire in Spanish too. It is possible that some authors just want to use your set of questions. We often do not reinvent the wheel when designing surveys and use questions from the World Values Surveys or the European Social Survey. Some surveys clearly undergo much more thinking when designed, but I cannot exclude that someone would use this one too for some purposes.

Although a bit unusual, we have now added the Spanish version of the questionnaire to the supplementary material as well.

2. References sometimes mix original studies and review papers, even if the review papers refer to the same original study. It may help the reader to refer to the original study for the finding and to the review paper for whatever additional intuition it provides, if any.

We consider it valuable to cite a review paper as it sums up findings from different studies. However, we have followed the reviewer's suggestion and adapted references where relevant.

3. It is not clear whether you designed 15 pilot questionnaires – then it would be very surprising that you used the current one – or whether you piloted your questionnaire on 15 respondents.

We piloted our questionnaire on 15 respondents. We have clarified this in the text.

4. The description of the Mokken scale in the main body of text and the data section is very unclear. It seems that your variables do not reach the minimum threshold of validity to be used. Then, you should not use them and switch to a different approach.

We understand your point and have now completely rewritten this text part in the Methods.

5. Sometimes the authors make comparisons without clarifying what the baseline case is. For instance: "Respondents considered returning revenues to poor households and the combination of this with supporting climate projects as the least effective in terms of emissions reductions." Compared to what?

We have clarified this example and double-checked other such instances.

References

Andersson, David, and Jonas Nässén. 2016. "The Gothenburg Congestion Charge Scheme: A Pre-Post Analysis of Commuting Behavior and Travel Satisfaction." *Journal of Transport Geography* 52: 82–89.

Baranzini, Andrea, Nicolas Borzykowski, and Stefano Carattini. 2018. "Carbon Offsets out of the Woods? Acceptability of Domestic vs. International Reforestation Programmes in the Lab." *Journal of Forest Economics* 32: 1–12.

Baranzini, Andrea, and Stefano Carattini. 2017. "Effectiveness, Earmarking and Labeling: Testing the Acceptability of Carbon Taxes with Survey Data." *Environmental Economics and Policy Studies* 19 (1): 197–227.

Baranzini, Andrea, Stefano Carattini, and Linda Tesauro. 2018. "The Geneva Congestion Charge: Rationale, Design, and Acceptability." *World Congress of Environmental and Resource Economists*.

Beiser-McGrath, Liam F., and Thomas Bernauer. 2019. "Could Revenue Recycling Make Effective Carbon Taxation Politically Feasible?" *Science Advances* 5 (9): eaax3323.

Carattini, Stefano, Andrea Baranzini, and Rafael Lalive. 2018. "Is Taxing Waste a Waste of Time? Evidence from a Supreme Court Decision." *Ecological Economics* 148: 131–51.

- Carattini, Stefano, Andrea Baranzini, Philippe Thalmann, Frédéric Varone, and Frank Vöhringer. 2017. "Green Taxes in a Post-Paris World: Are Millions of Nays Inevitable?" *Environmental and Resource Economics* 68 (1): 97–128.
- Carattini, Stefano, Maria Carvalho, and Sam Fankhauser. 2018. "Overcoming Public Resistance to Carbon Taxes." *Wiley Interdisciplinary Reviews: Climate Change* 9 (5): e531.
- Carattini, Stefano, Anomitra Chatterjee, Todd L. Cherry, and Steffen Kallbekken. 2018. "Climate Levy, Informational Messages, Acceptability, and Treatment Effects of Framing Effectiveness and Earmarking (CLIMATEFEE)." AEA RCT Registry. <https://doi.org/10.1257/rct.3476-1.0>.
- Carattini, Stefano, Steffen Kallbekken, and Anton Orlov. 2019. "How to Win Public Support for a Global Carbon Tax." *Nature* 565 (7739): 289–91.
- Cherry, Todd L., Steffen Kallbekken, and Stephan Kroll. 2014. "The Impact of Trial Runs on the Acceptability of Environmental Taxes: Experimental Evidence." *Resource and Energy Economics* 38 (C): 84–95.
- Douenne, Thomas, and Adrien Fabre. 2019. "Can We Reconcile French People with the Carbon Tax? Disentangling Beliefs from Preferences." 2019.10. Working Papers. FAERE - French Association of Environmental and Resource Economists. <https://ideas.repec.org/p/fae/wpaper/2019.10.html>.
- Hansla, André, Erik Hysing, Andreas Nilsson, and Johan Martinsson. 2017. "Explaining Voting Behavior in the Gothenburg Congestion Tax Referendum." *Transport Policy* 53: 98–106.
- Heres, David R., Steffen Kallbekken, and Ibon Galarraga. 2017. "The Role of Budgetary Information in the Preference for Externality-Correcting Subsidies over Taxes: A Lab Experiment on Public Support." *Environmental and Resource Economics* 66 (1): 1–15.
- INFRAS. 2015. "Klimaschutz Und Grüne Wirtschaft - Was Meint Die Bevölkerung? Ergebnisse Einer Repräsentativen Bevölkerungsbefragung - Schlussbericht." Zurich: Bundesamt für Umwelt.
- Marshall, George, Darragh Conway, Robin Webster, Louise Comeau, Daniel James Besley, and Isabel Saldarriaga Arango. 2018. "Guide to Communicating Carbon Pricing." 132534. The World Bank. <http://documents.worldbank.org/curated/en/668481543351717355/Guide-to-Communicating-Carbon-Pricing>.
- Schuitema, Geertje, Linda Steg, and Sonja Forward. 2010. "Explaining Differences in Acceptability before and Acceptance after the Implementation of a Congestion Charge in Stockholm." *Transportation Research Part A: Policy and Practice* 44 (2): 99–109.
- World Bank. 2019. "State and Trends of Carbon Pricing – 2019." Washington, DC.

Response to reviewer #2

This is a worthwhile topic that will be of interest to others. The methodology seems fine for the most part, and the sample is good. I think the paper is likely to be acceptable.

Thank you for your positive assessment.

But there are a few areas where a lack of clarity should be addressed. Especially, it is difficult to follow some of the reported results. Even the abstract could do a better job of summarizing the important information.

We have improved the abstract now.

One of the primary manipulations is presenting half the participants with information about the carbon tax. Did providing information actually increase either perceived or assessed knowledge? This would be worth assessing.

We agree that answering this question would be interesting. However, we can only compare the two groups regarding their acceptability level and perception of effectiveness and fairness as these were measured after information about carbon tax functioning was provided. But regarding their prior and perceived knowledge about the carbon tax we cannot do this, as we measured knowledge only before the information provision but not afterwards.

The second main independent variable is the use of the revenue from the carbon tax. It was not clear to me until I read the more detailed Methods that fairness and acceptability were rated both in general and for specific revenue uses. Are the general or specific ratings used in the data analyses? This is not clear.

We have now tried to make clearer from the start that this is the approach.

Individual fairness and individual wellbeing are not the same thing. They should not be conflated (e.g. line 79). Similarly, reducing the wellbeing of those who are worst off is not the same as increasing distributional fairness, though they are clearly related. In addition, describing these measures as measures of fairness makes it more confusing that you also measured what appear to be straightforward ratings of fairness. You should clarify this.

We understand the reviewer's concern and therefore no longer use the terms "Individual fairness" and "distributional fairness" when reporting our results, but "individual effects" and "low-income effects".

The figures are good.

Thank you.

I'm also confused by Table 2. The description refers to the third column twice, and I think the second time the fourth column is meant. But I'm not sure what the different analyses are that are being presented in the third and fourth column.

We have clarified this. The second time we referred to the third column should have mentioned the fourth column. This has now been corrected.

I spent some time going back and forth between the description of results and the description of the methods to try to better understand the analyses, but the reader should not have to work this hard. Please try to clarify the description of the results.

We have clarified the description of results. Please note, though, that the Nature journals require a Method section at the end of the paper, which makes the "going back and forth" that you describe unavoidable.

Reviewers' comments:

Reviewer #1 (Remarks to the Author):

The American Economic Association introduced pre-registration of randomized controlled trials, and gave them DOIs, to (i) increase transparency and prevent data mining; and (ii) recognize the fact that randomized controlled trial can take a lot of time to be conducted and authors' ideas need to be protected from others trying to run away with them to publish faster, even if with less thorough studies. Note also that the present paper no longer cites another (older) published article that used randomized information provision in the context of gasoline taxes, which appeared in the original submission (Kaplowitz and McCright 2015). Maybe the authors would argue that carbon taxes are very different from gasoline taxes. The authors also ignored a World-Bank-commissioned report on "communicating carbon pricing", despite the obvious relevance to their study (Marshall et al. 2018).

Also, the present paper arguably relates to other studies on Pigouvian taxes looking at belief revision between ex ante and ex post (Schuitema, Steg, and Forward 2010; Cherry, Kallbekken, and Kroll 2014; Andersson and Nässén 2016; Hansla et al. 2017; Carattini, Baranzini, and Lalive 2018). Information provision is, however, considered by the literature as a substitute, and possibly also a complement, to trial runs and actual experience, which themselves require public support to take place. The authors decided to ignore all these studies. In their response the authors also make the odd claim that the information that they provided to respondents did not lead to learning or belief revision. This is surprising, as it would mean that the main effect in their paper is driven only by priming and pro-social bias, which would go against the authors' conclusions. If that is what the authors consider to be happening, a detailed explanation in the paper would be needed.

Second, the authors introduced new tables comparing their sample with the underlying population, and the treated sub-sample with the control sub-sample, but these only include a subset of the variables that they have available (6 variables in the first case, 5 in the second one). The authors have several more socioeconomic characteristics to compare, as well as variables on (perceived) knowledge, which appear in the questionnaire before the randomized intervention. Of course, quotas used by marketing agencies to stratify when recruiting respondents do not exempt researchers to show representativeness also along dimensions other than those used to stratify, especially the most important ones for their study.

Third, the authors did not amend the manuscript to reflect or acknowledge the potential priming that the very odd ordering of questions in their paper likely generates. Their response is that priming is possible in all surveys – which is true, but most surveys ask about public support at the outset to minimize priming – and that addressing priming would require "a specialized research study". Of course, these design issues, which the authors acknowledge in the response to reviewers but not in the paper, were among the reasons for my definition of "poor design".

Fourth, the authors do not explain in the paper their choice for a continuous measure of acceptability, they simply include a rebuttal in the response to reviewers. A reader would likely wonder about it in the same way that I did. While some other studies, for instance in social psychology, did use continuous variables to measure public support, as the authors point out in their response, most studies use a dichotomous variable to mirror a ballot decision. Hence, an explanation is in order.

Fifth, the rationale for not including a tax rate is rather unconvincing. It may indeed be hard for lay people to understand the meaning of a \$40 per ton of CO₂ carbon tax, but that is precisely why the literature has provided indications in terms of price increases for common goods such as gas. A proper explanation is still in order.

Sixth, the authors keep considering questions that appear very late in the process, with respondents having to express their opinion on one use of revenues after the other. If authors in this literature tend to use split sample designs and focus only on the questions related to the split sample design is precisely because they consider that only a limited number of questions on the

very same issues can be credibly used. Bombarding respondents with various uses of revenues and repeating questions about acceptability and fairness for each of them is definitely not standard and a source of concern about the reliability of the resulting data. Plus, and related to the following point, one cannot really treat subsequent questions as variables independent from one another, as currently (still) done.

Reviewer #2 (Remarks to the Author):

The authors have satisfactorily addressed my concerns and I think the contributions of the manuscript are now apparent. However, there is still room for improving the clarity of the presentation of the results. I have three specific questions:

Line 16-17 is unclear. Is policy acceptability more strongly related with perceived fairness than policy acceptability is related to effectiveness, or more strongly than fairness is related to effectiveness?

Line 114 says "one possible explanation". Isn't this the only possible explanation, given that assessed knowledge was higher than perceived knowledge?

Line 154-155 describes the difference between the subsamples, but did you conduct an overall comparison, or only comparisons at each knowledge level?

Reviewer #3 (Remarks to the Author):

In this study, "Carbon tax acceptability with information provision and mixed revenue uses", Participants were given five options for carbon tax revenue: 1) return all revenue to low-income households, 2) support development of climate projects, 3) split revenues between option 1 and option 2, 4) redistribute revenues equally across all households, 5) split revenues between option 2 and 4. It's an interesting finding that people care more about fairness than effectiveness in the context of carbon tax, perhaps because effectiveness is diffuse, particularly to a sample that does not have a lot of exposure to the topic. It is also interesting that depending on how you ask people, they endorse different options for what ought to be done with the tax revenue. Information provision, the main experimental difference, appears to increase only acceptability but not perceptions of fairness.

Major concerns:

- How do we know the five questions asked to assess knowledge of carbon tax issues is objectively hard or easy? AKA where do these questions come from? I see no source, and the authors note that they removed the fifth item from the Mokken scale analysis due to low scalability statistics, but that the fifth item is included in the paper nevertheless (lines 406-425). So why run this analysis to begin with? Additionally, the Mokken scale analysis implemented only considers right or wrong answers. According to Figure 1A, the proportion of people that say they "don't know" ranges from 25% ("A carbon tax makes renewable energy sources, such as solar electricity, more expensive than fossil fuels") to nearly 50% ("A carbon tax mandates all producers of consumers which low-carbon technology they should adopt"). I question how "knowledge" is operationalized, in part because only a subset of responses (yes/no) were considered and in part because I don't know where the items are coming from, so cannot evaluate whether one would expect people to know.
- Survey is not counterbalanced (there are five stages that appear sequentially for all participants). No discussion of the inevitable order effects?
- Free expression of participants' preferred revenue allocation: do you think, given that there is no "genuine carbon tax" in Spain (line 70), the sample is invoking an ignorance prior, where absent any other information they endorse all possible options roughly equally? It seems these results may capture this phenomena rather than any stable preference for tax policies that your

population is admittedly not familiar with (line 70).

This seems like a likely alternative explanation for some of your results, particularly the tendency, when participants are given the choice, to split equally between all households and low-income households (right-most bar on all three graphs in Figure 5).

o Experimental work on ignorance prior (Fox and Clemen, 2005):
<https://pubsonline.informs.org/doi/abs/10.1287/mnsc.1050.0409>

- Line 347: One of the conditions for inclusion was if the quota they belonged to was already completed, of which responses for 123 participants were not included, or about 6% of participants of the total sample size you ended up including. I don't know what demographic variables these excluded participants are a part of. Have you done any analysis to ensure that these 123 participants do not significantly change your findings?

Minor concerns:

- In general, I have to read multiple sections more than twice to fully understand the gist of your argument and what tests you are running.

- I would have liked to see more theoretical summary rather than a re-statement of the results.

Nitpicky comments:

- Another recent study that looks at the effect of information provision in public support for carbon taxes finds also that when people learn more about carbon taxes, they are more likely to support such an initiative: <https://www.nature.com/articles/s41558-019-0474-0>

- Fig 2: the y-axes differ in range and increment for the self-perceived knowledge and assessed knowledge. Hard to do visual comparison.

o Why is the middle option the most disordinal for all three graphs, but particularly the assessed knowledge and the knowledge gap? What does it mean that those that rate carbon tax acceptability as a 3 (neither unacceptable nor acceptable, according to Appendix) have the lowest assessed knowledge and the highest knowledge gap?

- Shift between parametric and non-parametric statistics (e.g., z-score both measures in line 121) and then Mann-Whitney test (line 155-157). Distribution from Figure 2C looks non-parametric.

- Fig 4: y-axes differ in range and increment.

- Fig 5: y-axes differ in range and increment – would it be more suitable to express this in proportions rather than raw number counts, at least for the latter two figures (respondents with low acceptability and high acceptability)?

o Where do people who say they find the options unacceptable nor acceptable lie? Did you do a median split? Seems odd to do it any other way, but I don't see anything in the main text about how the middle category was divided up.

- Table 2

o What are these coefficients? Odds ratios? Where are the confidence intervals?

o Line 298: which pseudo-R-squared? Nagelkerke? MacFadden?

- Fig 3: Mann-Whitney test requires a Bonferroni correction as you are making multiple comparisons.

- There are lots of mentions of "weakly significance" (line 247) and negative interactions (line 259). In statistically significant interactions, I cannot tell by the sign alone how the interaction looks like (what goes up or down as a function of the other variable). I have no idea what these interactions that you discuss in lines 257-263 look like graphically.

Response to reviewer #1:

(Our response is in Italics)

The American Economic Association introduced pre-registration of randomized controlled trials, and gave them DOIs, to (i) increase transparency and prevent data mining; and (ii) recognize the fact that randomized controlled trial can take a lot of time to be conducted and authors' ideas need to be protected from others trying to run away with them to publish faster, even if with less thorough studies. Note also that the present paper no longer cites another (older) published article that used randomized information provision in the context of gasoline taxes, which appeared in the original submission (Kaplowitz and McCright 2015). Maybe the authors would argue that carbon taxes are very different from gasoline taxes. The authors also ignored a World-Bank-commissioned report on "communicating carbon pricing", despite the obvious relevance to their study (Marshall et al. 2018).

We have integrated the suggested literature in the Introduction.

Also, the present paper arguably relates to other studies on Pigouvian taxes looking at belief revision between ex ante and ex post (Schuitema, Steg, and Forward 2010; Cherry, Kallbekken, and Kroll 2014; Andersson and Nässén 2016; Hansla et al. 2017; Carattini, Baranzini, and Lalive 2018). Information provision is, however, considered by the literature as a substitute, and possibly also a complement, to trial runs and actual experience, which themselves require public support to take place. The authors decided to ignore all these studies. In their response the authors also make the odd claim that the information that they provided to respondents did not lead to learning or belief revision. This is surprising, as it would mean that the main effect in their paper is driven only by priming and pro-social bias, which would go against the authors' conclusions. If that is what the authors consider to be happening, a detailed explanation in the paper would be needed.

We have now included most of the suggested literature in the Introduction. Perhaps we have not been sufficiently clear in our first response. While we consider learning from a real-world trial run to be different from a simple information intervention in a survey, there is of course some overlap.

Second, the authors introduced new tables comparing their sample with the underlying population, and the treated sub-sample with the control sub-sample, but these only include a subset of the variables that they have available (6 variables in the first case, 5 in the second one). The authors have several more socioeconomic characteristics to compare, as well as variables on (perceived) knowledge, which appear in the questionnaire before the randomized intervention. Of course, quotas used by marketing agencies to stratify when recruiting respondents do not exempt researchers to show representativeness also along dimensions other than those used to stratify, especially the most important ones for their study.

In the original comment, the reviewer was not requesting comparison on all possible characteristics. Regarding the comparison with the treated sample, we used five obvious variables (climate concern, income, education, political orientation, political trust). We have now added further results for differences between treated and control subsample on assessed and self-perceived knowledge as well as car use and household size (see supplementary material). These are the variables we also used throughout our study as socioeconomic controls. However, the respective tests did not generate any statistically significant difference. With regard to comparing our sample to the population, we have done this for age, gender, income, education, household size and political views. Further tests are prohibited by lack of comparable data from the Spanish National Institute of Statistics.

Third, the authors did not amend the manuscript to reflect or acknowledge the potential priming that the very odd ordering of questions in their paper likely generates. Their response is that priming is possible in all surveys – which is true, but most surveys ask about public support at the outset to minimize priming – and that addressing priming would require “a specialized research study”. Of course, these design issues, which the authors acknowledge in the response to reviewers but not in the paper, were among the reasons for my definition of “poor design”.

We added a discussion of this limitation to the final section of the manuscript, which also relates to your sixth point below.

Fourth, the authors do not explain in the paper their choice for a continuous measure of acceptability, they simply include a rebuttal in the response to reviewers. A reader would likely wonder about it in the same way that I did. While some other studies, for instance in social psychology, did use continuous variables to measure public support, as the authors point out in their response, most studies use a dichotomous variable to mirror a ballot decision. Hence, an explanation is in order.

A 5-point scale evidently allows for more detailed analysis than a dichotomous scale. The latter might make sense for studies in countries where the political system follows such a procedure. In our research context, Spain, ballots seldom take place at the national level. We have added this explanation in the Methods section when discussing data collection.

Fifth, the rationale for not including a tax rate is rather unconvincing. It may indeed be hard for lay people to understand the meaning of a \$40 per ton of CO₂ carbon tax, but that is precisely why the literature has provided indications in terms of price increases for common goods such as gas. A proper explanation is still in order.

As we write in the Methods section “We did not include a tax rate as our aim was to inquire about peoples’ attitudes regarding carbon taxation in general, as a tool or mechanism, independent of a tax rate. People arguably have difficulty translating a concrete tax rate into a personal cost. Moreover, our aim was to obtain general results about how people perceive a carbon tax along with revenue uses, not affected by a specific rate”. While it is true that some studies provide information on the exact tax rate and how this is expected to alter prices of certain goods, many other studies (not only on carbon tax but also emission standards, etc.) ask about perceptions more generally without specifying features of policy design. See for instance the literature review of climate policy attitudes by Kysela et al. (2019), now included in the manuscript, which provides examples such as: “To what extent do you agree with the policy of putting a price on carbon?”. We hope that the reviewer thus accepts that it is not unusual to ask for opinions about a tax in general, without providing any rate. Consistent with our research aim, we were interested in respondents comparing different qualitative proposals.

Sixth, the authors keep considering questions that appear very late in the process, with respondents having to express their opinion on one use of revenues after the other. If authors in this literature tend to use split sample designs and focus only on the questions related to the split sample design is precisely because they consider that only a limited number of questions on the very same issues can be credibly used. Bombarding respondents with various uses of revenues and repeating questions about acceptability and fairness for each of them is definitely not standard and a source of concern about the reliability of the resulting data. Plus, and related to the following point, one cannot really treat subsequent questions as variables independent from one another, as currently (still) done.

This point is related to the previous one on potential order effects. We discuss both issues now in the revised manuscript. Please note, though, that our questionnaire's length of 15 minutes is far from being unusual.

Response to reviewer #2:

(Our response is in Italics)

The authors have satisfactorily addressed my concerns and I think the contributions of the manuscript are now apparent. However, there is still room for improving the clarity of the presentation of the results. I have three specific questions:

Line 16-17 is unclear. Is policy acceptability more strongly related with perceived fairness than policy acceptability is related to effectiveness, or more strongly than fairness is related to effectiveness?

We have clarified this in the manuscript now.

Line 114 says “one possible explanation”. Isn’t this the only possible explanation, given that assessed knowledge was higher than perceived knowledge?

We have reformulated the sentence.

Line 154-155 describes the difference between the subsamples, but did you conduct an overall comparison, or only comparisons at each knowledge level?

Thank you for the question. Indeed, in Table C.3 in the Appendix C we provide results of an ordered logit regression conducted on the whole sample to further support our conclusion. We have clarified this now.

Response to reviewer #3:

(Our response is in Italics)

In this study, “Carbon tax acceptability with information provision and mixed revenue uses”, Participants were given five options for carbon tax revenue: 1) return all revenue to low-income households, 2) support development of climate projects, 3) split revenues between option 1 and option 2, 4) redistribute revenues equally across all households, 5) split revenues between option 2 and 4. It’s an interesting finding that people care more about fairness than effectiveness in the context of carbon tax, perhaps because effectiveness is diffuse, particularly to a sample that does not have a lot of exposure to the topic. It is also interesting that depending on how you ask people, they endorse different options for what ought to be done with the tax revenue. Information provision, the main experimental difference, appears to increase only acceptability but not perceptions of fairness.

Thank you for this positive assessment.

Major concerns:

- How do we know the five questions asked to assess knowledge of carbon tax issues is objectively hard or easy? AKA where do these questions come from? I see no source, and the authors note that they removed the fifth item from the Mokken scale analysis due to low scalability statistics, but that the fifth item is included in the paper nevertheless (lines 406-425). So why run this analysis to begin with? Additionally, the Mokken scale analysis implemented only considers right or wrong answers. According to Figure 1A, the proportion of people that say they “don’t know” ranges from 25% (“A carbon tax makes renewable energy sources, such as solar electricity, more expensive than fossil fuels”) to nearly 50% (“A carbon tax mandates all producers of consumers which low-carbon technology they should adopt”). I question how “knowledge” is operationalized, in part because only a subset of responses (yes/no) were considered and in part because I don’t know where the items are coming from, so cannot evaluate whether one would expect people to know.

Given that there was no scale of carbon tax knowledge available in the literature, we developed it ourselves. We motivate better in the revised version the choice of these particular questions covering distinct parts of knowledge about the tax.

For reasons of transparency, we show the fifth item in Figure 1, but do not use it when constructing our knowledge scale (assessed knowledge) and running regression analyses. That is, the scale of the initial six items finally had five items, as we also state in the paper: “five items form a reliable one-dimensional scale”. In the Methods section we wrote that “In addition, neither the test of monotonicity nor intersection are violated in our case (even when keeping the fifth item). Finally, we measured the knowledge gap by maintaining all six items measuring assessed knowledge, i.e. not deleting item 5. Results are nearly identical, confirming the significant relation between carbon tax acceptability and the knowledge gap.” What we meant is that we also tried an alternative way of measurement and the results remained robust. We see this may confuse the reader, so we deleted this text now.

Regarding the use of dichotomous yes/no response, this procedure is in line with the cited literature on measuring climate knowledge using the Mokken scale (e.g. Tobler et al., 2012). Note also that we do not discard “don’t know” responses but in line with Tobler et al. consider them as false responses (i.e., 1 = “correct”, 0 = “wrong” and “don’t know”). We have clarified this in the revision.

- Survey is not counterbalanced (there are five stages that appear sequentially for all participants). No discussion of the inevitable order effects?

In line with comments by reviewer 1, we added a discussion of this point to the final section.

- Free expression of participants' preferred revenue allocation: do you think, given that there is no "genuine carbon tax" in Spain (line 70), the sample is invoking an ignorance prior, where absent any other information they endorse all possible options roughly equally? It seems these results may capture this phenomena rather than any stable preference for tax policies that your population is admittedly not familiar with (line 70).

This seems like a likely alternative explanation for some of your results, particularly the tendency, when participants are given the choice, to split equally between all households and low-income households (right-most bar on all three graphs in Figure 5).

o Experimental work on ignorance prior (Fox and Clemen, 2005): <https://pubsonline.informs.org/doi/abs/10.1287/mnsc.1050.0409>

Thank you for this suggestion. However, note that we show that respondents who preferred mixed revenue use composed of three uses do not endorse all three options (roughly) equally, but generally prefer to allocate a relatively large share of revenue to support climate projects. We further decided to not integrate your suggestion regarding prior ignorance as it would, in principle, also apply to results based on closed-ended survey questions. We feel this assumption about unstable preferences is too speculative at this point and would undercut some of our insights.

- Line 347: One of the conditions for inclusion was if the quota they belonged to was already completed, of which responses for 123 participants were not included, or about 6% of participants of the total sample size you ended up including. I don't know what demographic variables these excluded participants are a part of. Have you done any analysis to ensure that these 123 participants do not significantly change your findings?

These respondents were filtered out because the quota to which they belonged was already completed. Hence, this decision contributed exactly to the final sample being representative of the general population of Spain. This is a common procedure in web-based questionnaires implemented for a panel of respondents. We do not have data on the filtered-out respondents because the company that collected the data did not provide it to us, as it would make the sample non-representative of the general population. Hence, we also do not see the need to acquire and analyze these data, as it would not add relevant information to the study.

Minor concerns:

- In general, I have to read multiple sections more than twice to fully understand the gist of your argument and what tests you are running.

- I would have liked to see more theoretical summary rather than a re-statement of the results.

This may be partly due to the complexity of our study. All the co-authors have re-read the study carefully to address these points. The same holds for the valuable detailed comments below. We hope that through addressing these we have resolved the reviewer's difficulty to understand all the details of our study.

Nitpicky comments:

- Another recent study that looks at the effect of information provision in public support for carbon taxes finds also that when people learn more about carbon taxes, they are more likely to support such an initiative: <https://www.nature.com/articles/s41558-019-0474-0>

We thank the reviewer for this suggestion. We have included a reference to Hagmann et al. (2019) in the introduction when explaining the effects of information provision and communication on carbon tax acceptability. This paper shows the influence on policy support of the order in which policy options - carbon tax and green energy nudge - and their effectiveness are presented.

- Fig 2: the y-axes differ in range and increment for the self-perceived knowledge and assessed knowledge. Hard to do visual comparison.

We present the two plots using the same y-axis range now.

o Why is the middle option the most disordinal for all three graphs, but particularly the assessed knowledge and the knowledge gap? What does it mean that those that rate carbon tax acceptability as a 3 (neither unacceptable nor acceptable, according to Appendix) have the lowest assessed knowledge and the highest knowledge gap?

We have included a possible explanation now. Our results show that respondents who consider a carbon tax as “neither unacceptable nor acceptable” have on average the lowest assessed knowledge and self-perceived knowledge, though in the latter case the difference with other groups is not very large. It may mean that respondents having the least knowledge about the policy are least interested in the topic and therefore do not have an opinion about it.

- Shift between parametric and non-parametric statistics (e.g., z-score both measures in line 121) and then Mann-Whitney test (line 155-157). Distribution from Figure 2C looks non-parametric.

Here we use the procedure from Fernbach et al. (2019) “After z-scoring objective and self-assessed knowledge, we calculated a difference score by subtracting each participant’s objective knowledge score from their self-assessed knowledge score. This difference score, which represents gaps between self-assessed and objective knowledge, increases as extremity of opposition increases ($\beta_{\text{extremity}} = 28$; $t(499) = 8.77$; $P < 0.0001$; 95% CI (0.22, 0.35)).”

Hence, we normalize both variables, take their difference and then plot the result against carbon tax acceptability. We further add an OLS regression of knowledge gap on carbon tax acceptability.

- Fig 4: y-axes differ in range and increment.

We now use an identical range for the y-axis in all subplots of Figure 4.

- Fig 5: y-axes differ in range and increment – would it be more suitable to express this in proportions rather than raw number counts, at least for the latter two figures (respondents with low acceptability and high acceptability)?

We like this suggestion and now present the results in percentage terms, which together with the consistent y-axis allows for better comparability of the subplots.

o Where do people who say they find the options unacceptable nor acceptable lie? Did you do a median split? Seems odd to do it any other way, but I don’t see anything in the main text about how the middle category was divided up.

No, the middle group (respondents who choose “neither nor” option) have been excluded from the center and right charts of Figure 5. Thus, they are present only in the leftmost subplot of Figure 5. We clarify this in the revised manuscript now.

- Table 2

o What are these coefficients? Odds ratios? Where are the confidence intervals?

No, these are logs of the odds. We have now replaced them with odds ratios with confidence intervals and p-values.

o Line 298: which pseudo-R-squared? Nagelkerke? MacFadden?

It is Nagelkerke pseudo R2. We clarified that now.

- Fig 3: Mann-Whitney test requires a Bonferroni correction as you are making multiple comparisons.

Thank you, we replaced the Mann-Whitney test with Kruskal test with Bonferroni correction. The results remain virtually the same. We clarify this now in the paper.

- There are lots of mentions of “weakly significance” (line 247) and negative interactions (line 259). In statistically significant interactions, I cannot tell by the sign alone how the interaction looks like (what goes up or down as a function of the other variable). I have no idea what these interactions that you discuss in lines 257-263 look like graphically.

In Table 1 we have two types of interaction effects: revenue uses with information provision and revenue uses with assessed knowledge. When we interpret the coefficient of the interaction effect with information provision, it becomes obvious that a positive and significant effect means that people who received additional information about the tax chose a higher value of the dependent variable (be it acceptability of a carbon tax or perceived effectiveness or fairness) for the given revenue use option. Table C.4 in the Supplementary materials supports this as there we run regressions for each revenue use separately and use the dummy variable on information provision as one of the explanatory variables. As you can see, the results remain robust.

When we interpret a positive/negative coefficient of an interaction effect with the assessed knowledge, the same logic holds. Thus, when we write “The interaction is negative, however, with transfers to low-income households.” It means that the more knowledge people have on carbon tax, the more they tend to dislike a carbon tax given its revenue will be distributed to low-income households. Again, Table C.4 in the Supplementary materials running regressions separately for each revenue use supports this intuition. We revised the respective paragraph to clarify these issues.

REVIEWER COMMENTS

Reviewer #2 (Remarks to the Author):

This seems to be an interesting and valuable paper.

Reviewer #3 (Remarks to the Author):

My primary concern relates to the wholesale rejection of my proposed alternative explanation of some of the results in this paper. I proposed previously that rather than reflecting any true stable preference about a topic that people are self-admittedly unfamiliar with, that people in this sample are invoking a well-documented behavioral bias called the ignorance prior by endorsing the three options roughly equally when given the option.

My original comment:

“Free expression of participants’ preferred revenue allocation: do you think, given that there is no “genuine carbon tax” in Spain (line 70), the sample is invoking an ignorance prior, where absent any other information they endorse all possible options roughly equally? It seems these results may capture this phenomena rather than any stable preference for tax policies that your population is admittedly not familiar with (line 70). This seems like a likely alternative explanation for some of your results, particularly the tendency, when participants are given the choice, to split equally between all households and low-income households (right-most bar on all three graphs in Figure 5). o Experimental work on ignorance prior (Fox and Clemen, 2005): <https://pubsonline.informs.org/doi/abs/10.1287/mnsc.1050.0409>”

The authors’ response is this:

“Thank you for this suggestion. However, note that we show that respondents who preferred mixed revenue use composed of three uses do not endorse all three options (roughly) equally, but generally prefer to allocate a relatively large share of revenue to support climate projects. We further decided to not integrate your suggestion regarding prior ignorance as it would, in principle, also apply to results based on closed-ended survey questions. We feel this assumption about unstable preferences is too speculative at this point and would undercut some of our insights.”

If I understand correctly, the authors’ state two reasons for their rebuttal:

- 1) the proportions are NOT roughly equal, and participants choose to endorse more revenue to climate projects, and
- 2) because this alternative explanation would, “in principle, also apply to results based on the close-ended survey questions” and are “too speculative”.

Here is my response to these points:

For point 1:

If it is the case that the proportions are not roughly equal, I would like to see a statistical test to corroborate this. A quick chi-square test of the two proportions the participants reported (41 vs 50%, line 220) is non-significant, chi-squared statistic $X^2 = 0.89$, $df = 1$, $p = 0.35$. Because I did not know the sample sizes of people who supported the carbon tax and those who didn’t, I could not include this into the test (and it is possible that a z test is more appropriate here to compare proportions directly, but again, that requires a sample size).

I also don’t know what the endorsed preferences of allocations are for those who chose the median

option (and how many people are in this category), and thus were excluded from all three figures of Figure 5. That would be an interesting comparison to see that, compared with those who support and do not support the carbon tax, people who chose the median option (neither supported nor supported the tax) allocated their revenue use differently.

An additional test to conduct may be to compare the proportion of endorsement of the three options across the groups (those that did not support carbon tax, those that did, and potentially those that chose the median option) for the participants who did endorse all three options (the far-most-right stacked bar in center and right plots in Figure 5). My admittedly unprecise guesstimates of these proportions from a visual scan (center plot = 24% for Climate, 17% for PoorHH, 17% AllHH; right plot = 31% Climate, 17% PoorHH, 14% AllHH) yields a chi-square statistic of $X^2 = 1.05$, $df = 2$, $p = 0.59$. Again, these exclude sample sizes and could be updated.

Finally, I have only considered the situation where participants see all three options. If these tests differ when there are two options presented (Climate compared with AllHH vs. Climate compared with PoorHH), that would also provide further evidence that people's preferences are not uniformly stable in favor of climate projects, in contrast to what the paper concludes (line 317-319). If this is the case, a potential reason could be because, at least in the case of the survey in this paper, there are arguably two social goods competing for revenue: climate projects as well as a redistribution of revenue to (low-income) households.

For point 2:

In addition to there being "no genuine carbon tax" in Spain (line 74), implying little or real-life exposure to this topic for the typical Spanish resident, I believe that there is already sufficient empirical evidence that the participants are unfamiliar with carbon tax knowledge, which would provide a reason for endorsing choices equiprobably when given the option to.

Empirically, in the six-item question survey that assesses knowledge about a carbon tax, the percentage of people who explicitly say they don't know ranges from 25-50% across the six items (Figure 1a). When asked to assess their self-perceived knowledge, over 1000 participants say "not at all" and another 600+ participants report "a little" (Figure 1b). Given that the $N = 2534$, that's at least 63% of participants who say they have no or low knowledge about a carbon tax. So it seems strange and inaccurate to say that this hypothesis of an ignorance prior is "speculative", when well more than half of people in your study admit to not knowing much, or anything, about carbon taxes.

A general comment:

Even if there are no differences in the proportion of endorsement, I still think there is a story here: about the importance of information provision in climate change education and the various dimensions of public support required to better understand how best to garner and increase public support of carbon emission reductions and other environmentally-friendly policies. These dimensions appear to include not just effectiveness but also other-regarding preferences such as equity (perceived fairness) and, to a lesser extent, acceptability. In fact, it seems that people's perceptions may be malleable (or able to be made malleable) which would support existing research on how inconsistent judgments can be related to perceptions of climate change.

In the introduction, the authors point out astutely that the public generally does not understand that the driving force of carbon taxes, rather than a concerted effort to funnel these revenues into other government projects, is ultimately to reduce emissions (a social good). Overall, the paper seems to provide evidence supporting information provision interventions that ultimately work towards linking the public's perceptions of carbon taxation with potentially non-monetary, other-regarding preferences, rather than the conventional appeal to the rational Homo Economicus man.

Response to the reviewers

Response to reviewer #2:

(Our response is in Italics)

This seems to be an interesting and valuable paper.

Thank you for this positive assessment.

Note that during the revision we discovered that the order of two options of revenue use, namely allocation to all households and mixed allocation to low-income households and climate projects, was reverse in the coding from that in the questionnaire. As a result, the interpretations were also reversed in the text. We have now corrected this.

Response to reviewer #3:

(Our response is in Italics)

My primary concern relates to the wholesale rejection of my proposed alternative explanation of some of the results in this paper. I proposed previously that rather than reflecting any true stable preference about a topic that people are self-admittedly unfamiliar with, that people in this sample are invoking a well-documented behavioral bias called the ignorance prior by endorsing the three options roughly equally when given the option.

My original comment:

“Free expression of participants’ preferred revenue allocation: do you think, given that there is no “genuine carbon tax” in Spain (line 70), the sample is invoking an ignorance prior, where absent any other information they endorse all possible options roughly equally? It seems these results may capture this phenomena rather than any stable preference for tax policies that your population is admittedly not familiar with (line 70). This seems like a likely alternative explanation for some of your results, particularly the tendency, when participants are given the choice, to split equally between all households and low-income households (right-most bar on all three graphs in Figure 5). o Experimental work on ignorance prior (Fox and Clemen, 2005): <https://pubsonline.informs.org/doi/abs/10.1287/mnsc.1050.0409> “

The authors’ response is this:

“Thank you for this suggestion. However, note that we show that respondents who preferred mixed revenue use composed of three uses do not endorse all three options (roughly) equally, but generally prefer to allocate a relatively large share of revenue to support climate projects. We further decided to not integrate your suggestion regarding prior ignorance as it would, in principle, also apply to results based on closed-ended survey questions. We feel this assumption about unstable preferences is too speculative at this point and would undercut some of our insights.”

If I understand correctly, the authors’ state two reasons for their rebuttal:

- 1) the proportions are NOT roughly equal, and participants choose to endorse more revenue to climate projects, and
- 2) because this alternative explanation would, “in principle, also apply to results based on the close-ended survey questions” and are “too speculative”.

Here is my response to these points:

For point 1:

If it is the case that the proportions are not roughly equal, I would like to see a statistical test to corroborate this. A quick chi-square test of the two proportions the participants reported (41 vs 50%, line 220) is non-significant, chi-squared statistic $X^2 = 0.89$, $df = 1$, $p = 0.35$. Because I did not know the sample sizes of people who supported the carbon tax and those who didn't, I could not include this into the test (and it is possible that a z test is more appropriate here to compare proportions directly, but again, that requires a sample size).

We have now included a new Figure C.3 in Appendix C of the Supplementary Material which shows the results of analysing individual-level data about the allocation by respondents of carbon-tax revenues between three options. It shows that for both samples (i.e. the complete sample with 2004 respondents and the sub-sample with 1231 respondents who preferred a mix of all three revenue uses) the average share of revenues allocated to climate projects is higher than the average shares for the two alternative options.

To formally test if the share of money allocated to climate projects exceeds that allocated to the other two options, we ran pairwise Mann-Whitney tests, shown in Figure C.4. in Appendix C. The results confirm that the difference is statistically significant.

I also don't know what the endorsed preferences of allocations are for those who chose the median option (and how many people are in this category), and thus were excluded from all three figures of Figure 5. That would be an interesting comparison to see that, compared with those who support and do not support the carbon tax, people who chose the median option (neither supported nor supported the tax) allocated their revenue use differently.

We believe there is some misunderstanding about Figure 5 as the left chart depicts all respondents, including those who chose the median option. In other words, respondents who chose the median option (3 on the Likert scale) were only excluded from the central and right charts of the figure, as is clarified in the footnote to Figure 5.

We undertook the analysis only for respondents who chose the median option and find that the distribution of revenues for this group is very similar to that of the total population. We now mention this additional analysis in the main text and added a new graph to the Appendix (see Figure C.2 in Appendix C).

An additional test to conduct may be to compare the proportion of endorsement of the three options across the groups (those that did not support carbon tax, those that did, and potentially those that chose the median option) for the participants who did endorse all three options (the far-most-right stacked bar in center and right plots in Figure 5). My admittedly unprecise guesstimates of these proportions from a visual scan (center plot = 24% for Climate, 17% for PoorHH, 17% AllHH; right plot = 31% Climate, 17% PoorHH, 14% AllHH) yields a chi-square statistic of $X^2 = 1.05$, $df = 2$, $p = 0.59$. Again, these exclude sample sizes and could be updated.

Please note that the percentages on the y-axis of Figure 5 are proportions of the overall sample and do not represent shares of revenue uses chosen by the respondents. Therefore, all the bars sum up to 100. Hence, the shares of climate which you estimated for the central and rightmost chart are not 24% and 31% but 41% and 50%.

Regarding your main concern, we interpret it as that you would like to see a test of whether allocation of revenues for climate projects varied with the acceptability of the policy among respondents who endorsed all three revenue options. If we just look at the means, those who rejected a carbon tax allocated on average 40.74%, those indifferent 41.42% and those who supported the tax 49.79%. Applying a Kruskal-Wallis rank sum test to compare the allocation of revenues to climate projects between these three groups, we find a chi-squared = 58.08 with $df = 2$ and p -value = 0. This means that the allocation of revenues significantly differs between the three groups. More specifically, while the choices of those who reject and of those who are indifferent about a carbon tax are not significantly different (p -value = 0.82), carbon-tax supporters make choices that are significantly different from both those who reject (p -value < 0.0001) and those who are indifferent (p -value < 0.0001). We have now added this information in the manuscript.

Finally, I have only considered the situation where participants see all three options. If these tests differ when there are two options presented (Climate compared with AllHH vs. Climate compared with PoorHH), that would also provide further evidence that people's preferences are not uniformly stable in favor of climate projects, in contrast to what the paper concludes (line 317-319). If this is the case, a potential reason could be because, at least in the case of the survey in this paper, there are arguably two social goods competing for revenue: climate projects as well as a redistribution of revenue to (low-income) households.

This is an interesting suggestion. However, by design, we presented all three revenue use options to respondents. Therefore, we cannot analyse now what would happen when only two options are presented.

For point 2:

In addition to there being “no genuine carbon tax” in Spain (line 74), implying little or real-life exposure to this topic for the typical Spanish resident, I believe that there is already sufficient empirical evidence that the participants are unfamiliar with carbon tax knowledge, which would provide a reason for endorsing choices equiprobably when given the option to.

Empirically, in the six-item question survey that assesses knowledge about a carbon tax, the percentage of people who explicitly say they don't know ranges from 25-50% across the six items (Figure 1a). When asked to assess their self-perceived knowledge, over 1000 participants say “not at all” and another 600+ participants report “a little” (Figure 1b). Given that the $N = 2534$, that's at least 63% of participants who say they have no or low knowledge about a carbon tax. So it seems strange and inaccurate to say that this hypothesis of an ignorance prior is “speculative”, when well more than half of people in your study admit to not knowing much, or anything, about carbon taxes.

Indeed, our results indicate that participants have low self-reported carbon-tax knowledge. However, Figures 2a/b also show that most people provide at least some correct responses to our set of knowledge questions. They suggest that assessed knowledge may be higher than self-reported one. People may be uncertain about their own knowledge. In any case, one cannot conclude that there is an “ignorance-prior” effect, as we have shown above that respondents do not endorse all possible options (roughly) equally. But we agree with the thrust of your idea and hence mention the possibility of such an effect now on page 9.

A general comment:

Even if there are no differences in the proportion of endorsement, I still think there is a story here: about the importance of information provision in climate change education and the various dimensions of public support required to better understand how best to garner and increase public support of carbon emission reductions and other environmentally-friendly policies. These dimensions appear to include not just effectiveness but also other-regarding preferences such as equity (perceived fairness) and, to a lesser extent, acceptability. In fact, it seems that people's perceptions may be malleable (or able to be made malleable) which would support existing research on how inconsistent judgments can be related to perceptions of climate change.

In the introduction, the authors point out astutely that the public generally does not understand that the driving force of carbon taxes, rather than a concerted effort to funnel these revenues into other government projects, is ultimately to reduce emissions (a social good). Overall, the paper seems to provide evidence supporting information provision interventions that ultimately work towards linking the public's perceptions of carbon taxation with potentially non-monetary, other-regarding preferences, rather than the conventional appeal to the rational Homo Economicus man.

We thank the reviewer for these positive considerations. We have tried to make them come out more clearly now in the conclusions.

Note that during the revision we discovered that the order of two options of revenue use, namely allocation to all households and mixed allocation to low-income households and climate projects, was reverse in the coding from that in the questionnaire. As a result, the interpretations were also reversed in the text. We have now corrected this.

REVIEWER COMMENTS

Reviewer #3 (Remarks to the Author):

Comments to authors:

- The first comment is that the authors say they use a Kruskal-Wallis rank sum test (line 237) to compare the allocation of revenues across three distinct options. A Kruskal-Wallis test measures whether three or more populations have equal mean ranks on some same outcome variable. Here there are one population (those who rejected carbon tax) who made three decisions (to climate projects, those indifferent to where revenue would go, and those who supported carbon tax). The authors report a chi-square test. Which one did they report?

- The authors note they found a coding error in one of their primary dependent variables: that reversed allocation to all households vs. to that of low-income households and climate projects and thus many of their interpretations were changed. This discovery was only found out after my repeated urging to run a simple chi-square test (which the authors possibly conducted, see comment 1). I find this perturbing and makes me wonder what other hidden errors are in the manuscript, and ultimately gives me little faith in the scientific validity of the manuscript, particularly in light of other statistical and experimental issues that I and other reviewers have commented on over the course of this revision process.

Response to reviewer #3:

(Our response is in Italics)

- The first comment is that the authors say they use a Kruskal-Wallis rank sum test (line 237) to compare the allocation of revenues across three distinct options. A Kruskal-Wallis test measures whether three or more populations have equal mean ranks on some same outcome variable. Here there are one population (those who rejected carbon tax) who made three decisions (to climate projects, those indifferent to where revenue would go, and those who supported carbon tax). The authors report a chi-square test. Which one did they report?

As we write in the paper, we use the Kruskal-Wallis rank sum test. This examines the equality of allocation of revenues to climate projects between people who rejected the tax (value 1 in image below), are indifferent (value 3) and supported the tax (value 5). Hence, the test is applied to three groups (or “sub-populations”), and not to one “population”. Note further that “chi-square” is the H-statistic of the Kruskal–Wallis test, which is approximately chi-square distributed (see also the terminology in the image below).

```
> conover.test(data_selected[,1], g=data_selected[,2],method = "bonferroni",alpha=0.01)
Kruskal-wallis rank sum test

data: x and group
kruskal-wallis chi-squared = 58.0752, df = 2, p-value = 0

Comparison of x by group
(Bonferroni)

Col Mean |
Row Mean |      1      3
-----|-----
  3 | -0.598735  0.8242
    |
  5 | -7.175856 -5.606275
    |      0.0000*  0.0000*

alpha = 0.01
Reject Ho if p <= alpha/2
```

We report the Kruskal-Wallis test rejecting the null hypothesis that allocation of revenues to climate is the same across people with different levels of support (“a chi-squared = 58.08 with $df = 2$ and p -value = 0”). In addition, we report p -values from comparing every possible pair of revenue options. In the text we wrote “More specifically, while choices by those who reject and by those who are indifferent about a carbon tax are not significantly different (p -value = 0.82), choices by carbon-tax supporters are significantly different from both those who reject (p -value < 0.0001) and those who are indifferent (p -value < 0.0001).”

- The authors note they found a coding error in one of their primary dependent variables: that reversed allocation to all households vs. to that of low-income households and climate projects and thus many of their interpretations were changed. This discovery was only found out after my repeated urging to run a simple chi-square test (which the authors possibly conducted, see comment 1). I find this perturbing and makes me wonder what other hidden errors are in the manuscript, and ultimately gives me little faith in the scientific validity of the manuscript, particularly in light of other statistical and experimental issues that I and other reviewers have commented on over the course of this revision process.

We would like to moderate your statement that “many of their interpretations were changed”. In fact, just a few interpretations were adapted. More importantly, the mistake did not change the conclusions related to one of our main research questions, namely that revenue mixes do not lead to significantly higher policy acceptance compared to single revenue uses.

In addition, we have been open and transparent with you and the journal, and once the mistake was found, we corrected and reported it to you. Please note that the mistake is not due to coding

or statistical errors, but a consequence of miscommunication by the survey company about the order of columns in the original raw data file. Nevertheless, we carefully checked the entire code and data which did not deliver any errors. The full code (in R software) is appended to the manuscript and thus open to review.

Finally, your suggestion that you “repeatedly” requested for a chi-square test strikes us as unfair. To our knowledge, this point was only once mentioned, namely in the previous revision round, and we immediately and constructively responded to it.

We hope you are satisfied by these clarifications. We did not see any need to make further changes in the manuscript.